# Why does a conceptual hydrological model fail to correctly predict discharge changes in response to climate change?

Doris Duethmann[1, 2], Günter Blöschl[1], Juraj Parajka[1]

[1]Institute for Hydraulic and Water Resources Engineering, Vienna University of Technology, Karlsplatz 13/223, 1040 Vienna, Austria.
[2]IGB Leibniz-Institute of Freshwater Ecology and Inland Fisheries, Müggelseedamm 310, 12587 Berlin, Germany.

*Correspondence to*: Doris Duethmann (duethmann@igb-berlin.de)

**Abstract.** Several studies have shown that hydrological models do not perform well when applied to periods with climate conditions that differ from those during model calibration. This has important implications for the application of these models in climate change impact studies. The causes of the low transferability to changed climate conditions have, however, only been investigated in a few studies. Here we revisit a study in Austria that demonstrated the inability of a conceptual semi-distributed HBV-type model to simulate the observed discharge response to increases in precipitation and air temperature. The aim of the paper is to shed light on the reasons of these model problems. We set up hypotheses for the possible causes of the mismatch between the observed and simulated changes in discharge and evaluate these using simulations with modifications of the model. In the baseline model, trends of simulated and observed discharge over 1978−2013 differ, on average over all 156 catchments, by $95 \pm 50$ mm yr$^{-1}$ per 35 yrs. Accounting for variations in vegetation dynamics, as derived from a satellite-based vegetation index, in the calculation of reference evaporation explains $36 \pm 9$ mm yr$^{-1}$ per 35 yrs of the differences between the trends in simulated and observed discharge. Inhomogeneities in the precipitation data, caused by a variable number of stations, explain $39 \pm 26$ mm yr$^{-1}$ per 35 yrs of this difference. Extending the calibration period from 5 to 25 yrs, including annually aggregated discharge data or snow cover data in the objective function, or estimating evaporation with the Penman-Monteith instead of the Blaney-Criddle approach, has little influence on the simulated discharge trends (5 mm yr$^{-1}$ per 35 yrs or less). The precipitation data problem highlights the importance of using precipitation data based on a stationary input station network when studying hydrologic changes. The model structure problem with respect to vegetation dynamics is likely relevant for a wide spectrum of regions in a transient climate and has important implications for climate change impact studies.

# 1    Introduction

A vast number of studies employ hydrological models to estimate climate change impacts on hydrology. In these studies, hydrological models are typically calibrated in the present climate and then run with climate input derived from climate models. However, hydrological predictions under changed climatic conditions are challenging as it is not clear whether the current generation of hydrologic models performs well under change (Blöschl and Montanari, 2010). By definition, testing models under future climate conditions is not possible, as future observations are not available. However, climatic changes have already been observed in the last decades. Hindcast simulations during periods with climatic variations in the past allow testing the suitability of hydrological models under changing climatic conditions. In the differential split sample test (DSST), suggested by Klemeš (1986), a hydrological model is evaluated in a period with climate conditions that differ from those during calibration. Though climatic contrasts between current and future conditions are likely larger than those in the observed record and future conditions will involve higher air temperatures and higher atmospheric $CO_2$ concentrations, further increasing uncertainties (Stephens et al., 2020), passing the DSST can be seen as a minimum requirement for models applied in climate impact assessments.

Studies that investigated the performance of hydrological models this way, evaluating them in periods with climatic conditions that differ from those of the model calibration, largely found a decrease in model performance (Seibert, 2003; Vaze et al., 2010; Merz et al., 2011; Coron et al., 2012; Seiller et al., 2012). In a study on four catchments in Sweden, large flood peaks in the evaluation period were strongly underestimated by the HBV model if the calibration period only contained small flood peaks (Seibert, 2003). Vaze et al. (2010) analysed the model performance of four lumped hydrological models in 61 catchments in southeast Australia when the model was calibrated to selected wet or dry periods of variable length. The reductions in model performance were greater with increasing difference in rainfall between calibration and evaluation periods. While most studies report reduced model performance in contrasting climates, Vormoor et al. (2018) did not find reduced model performance under contrasting conditions in terms of flood seasonality and flood generating processes, when applying a conceptual hydrological model in five catchments with changes in flood seasonality and flood generating processes in Norway.

Low model performance in contrasting climates is often characterized by biased discharge values (Coron et al., 2014; Kling et al., 2015). This is a serious concern since changes in discharge volume are of high interest in climate change impact studies. Merz et al. (2011) calibrated and evaluated the HBV model in 5-year periods in 273 catchments in Austria. They found that median flows were overestimated by 15 % and high flows by 35 % when parameters calibrated during 1976–1981 were applied to 2001–2006. Several studies found increased differences in discharge bias between the calibration and evaluation period with increasing differences in precipitation (Coron et al., 2012; Sleziak et al., 2018).

The problem of poor model performance in contrasting climates has been observed for various model structures. While most studies that investigate the transferability of hydrological models focus on lumped conceptual models, low transferability in contrasting climate has also been observed for semi-distributed conceptual models (Merz et al., 2011; Coron et al., 2014) and

process-based models (Magand et al., 2015). The application of a DSST to three different lumped conceptual models in five catchments in Tunisia showed similar problems of model transferability under contrasting climate conditions for the three models (Dakhlaoui et al., 2017). Seiller et al. (2012) tested the transposability of 20 lumped conceptual hydrological models between periods with contrasting precipitation and air temperature for two catchments in Canada and Germany and they were not able to
identify a specific model structure that performed well in contrasting climate for all their test conditions.

Understanding the causes of poor performance in a transient climate is a key question since this determines the way forward for hydrological modelling in a transient climate. Possible causes include data problems, poor parameterization of the model, or structural inadequacy (Coron et al., 2014; Westra et al., 2014; Fowler et al., 2018). In case of data problems, the model should be calibrated with corrected data; however, apart from this, simulations with projections of future climate should not be affected by
this problem. In case of parameterization problems, efforts should be invested in choosing calibration methods that result in reliable parameterizations in a transient climate. If the problem is related to the model structure, it will be important to understand what parts of the model structure result in reduced performance in order to avoid these structural components in climate change impact analyses. An example of data problems that may cause poor model performance under contrasting climate conditions are inhomogeneities in the precipitation data, which lead to biased estimates of the precipitation changes. Such inhomogeneities may
be caused by inhomogeneities in the station data itself, a variable number of stations included in a gridded data set (Fawcett et al., 2010), or climate variations that lead to changes in the undercatch error (Forland and Hanssen-Bauer, 2000). A poor parameterization may be caused by a too short calibration period. However, in several studies that observed poor performance in contrasting climate the problem could not be solved by using a longer calibration period (Luo et al., 2012; Brigode et al., 2013; Coron et al., 2014). Too low sensitivity of the objective function to the long-term dynamics of discharge may be another cause for
a poor parameterization that results in poor performance in a transient climate. Hartmann and Bárdossy (2005) observed increased transferability of a distributed conceptual hydrological model under contrasting climate conditions when including annually aggregated discharge data in the objective function in addition to daily discharge data. A thorough approach to test whether the problem may be solved by improving the parameterization is by applying multiobjective calibration to the different periods with contrasting climate (Fowler et al., 2018). Model structural inadequacy in the context of a transient climate includes changes in
catchment characteristics or dominant hydrological processes that are not reflected by the model. For example, changes in the glacier volume or a longer vegetation period may alter the hydrologic response of the catchment and result in deviations between simulated and observed discharge if not accounted for in the model. Despite their relevance for hydrological modelling in a transient climate, the causes of poor performance under contrasting climate conditions have only been investigated in a few studies (Westra et al., 2014; Fowler et al., 2016; Fowler et al., 2018).

This study aims at contributing to closing this gap by analysing the causes of the poor performance of a hydrological model in a transient climate for a case study on a large number of catchments in Austria. Due to a strong climate signal over the last decades (Schöner et al., 2011), Austria is well suited for studying climate-induced hydrologic changes. We applied a semi-distributed

hydrological model based on the HBV concept, which is widely used for operational and scientific purposes including climate impact assessments. However, in the study by Merz et al. (2011) (Merz2011 in the following), the model was not able to correctly estimate changes in mean discharge in response to the observed increases in precipitation and air temperature. Applying the model calibrated during 1976–1981 with climate data of 2001–2006 resulted in an increase of simulated discharge of on average 15 %, whereas observations show relatively stable annual discharge volumes. Here, we revisit the study by Merz2011 and investigate what causes the differences between simulated and observed changes in discharge. For that purpose, we set up hypotheses that are tested using modifications of the model. In particular, we analyse the effect of varying the input data for precipitation and air temperature, increasing the length of the calibration period, including annually aggregated discharge data or snow cover data in the objective function, and varying the calculation of reference evaporation ($E_{ref}$) to consider changes in global radiation and vapour pressure as well as changes in vegetation dynamics.

## 2 Data and methods

### 2.1 Study area

This study was carried out using data from 156 catchments in Austria (Figure 1). The catchments were selected based on the availability of daily discharge data for 1977–2014 (hydrological years, November to October; maximum of two years missing). We generally excluded catchments with substantial anthropogenic influences from dams or water withdrawals (Viglione et al., 2013), glaciers, and catchments where discharge exceeded the precipitation estimate. The more rigorous selection resulted in a smaller set of catchments compared to Merz2011, who used a set of 273 catchments. The median (interquartile range) of the catchment sizes is 192 (95/366) km². The data set includes lowland and mountain catchments and the median elevation range is 520 (372/665)–1593 (984/2126) m, (numbers in brackets refer to the interquartile range). The most frequent land cover is forest, which covers on average 52(40/67) % of the catchment area (based on Corine 2000 data; European Environment Agency (2016)), and grassland, which covers 23(14/33) % of the catchment area. In most catchments the fraction of arable land and heterogeneous agricultural areas is small with a median of 5(0/29) % of the catchment area. The study region shows strong climatic changes over the recent decades. On average over the study catchments, annual precipitation increased by $32 \pm 23$ mm yr$^{-1}$ or $2.4 \pm 1.7$ % per decade, air temperature increased by $0.45 \pm 0.09$ °C per decade and global radiation increased by $5.1 \pm 0.9$ W m$^{-2}$ per decade over the period 1977–2014. In contrast, discharge did not show strong trends and the average trend over the study period was $0.2 \pm 3.1$ % per decade (Duethmann and Blöschl, 2018).

### 2.2 Hydrometeorological data

Discharge data were provided by the Central Hydrographical Bureau (HZB) in Vienna. Climate data required by the hydrological model are air temperature, precipitation, and, depending on the model variant, relative humidity, global radiation and wind speeds.

Furthermore, interpolated snow depth data were used for model calibration in one model variant. The baseline precipitation data set (P0) was derived by spatially interpolating daily precipitation values of the available stations from HZB and the Austrian Central Institute for Meteorology and Geodynamics (ZAMG) using external drift kriging with elevation as auxiliary variable to a 1 km$^2$ grid, as in Merz2011. Due to variations in the station network, the number of stations included in the interpolation varies over time. In addition, two alternative precipitation data sets were used. As the first alternative (P1), we used the gridded SPARTACUS data set (Hiebl and Frei, 2018). It has a temporal and spatial resolution of 24 h and 1 km and is based on a two-step interpolation scheme. In the first step, a monthly background climatology for 1977–2006 was obtained based on 1249 stations (including 119 totalizer precipitation gauges), and in the second step, a constant number of 523 stations was used for interpolating ratios between the daily precipitation and the background climatology. For the second alternative precipitation data set (P2), we added a correction for systematic underestimation from gauge undercatch to the SPARTACUS data set using the following equation (Richter, 1995)

$$P_{\text{corr}} = P_{\text{orig}} + b \cdot P_{\text{orig}}{}^{e} \tag{1}$$

where $P_{\text{corr}}$ is undercatch corrected precipitation, $P_{\text{orig}}$ uncorrected precipitation, and $b$, $e$ are coefficients that depend on season, precipitation type and wind exposure. We estimated the precipitation type as snow for mean air temperatures below −1°C, as mixed precipitation between −1°C and 3°C, and as rain for mean air temperatures above 3°C (ATV-DVWK, 2001). The coefficients of Richter (1995) for very sheltered locations were applied to all grid points. On average over all catchments, the undercatch correction increased precipitation by 7.2 % compared to the original data without undercatch correction.

The baseline data set for mean daily air temperature (T0) was derived by spatially interpolating mean daily air temperatures of the available stations from the ZAMG using local ordinary least-squares regression with elevation, as in Merz2011. In addition, we used the gridded SPARTACUS data set (Hiebl and Frei, 2016), which is based on a constant station network of 150 stations, as alternative input (T1). Air temperature and precipitation were aggregated to averages by elevation zone for each catchment, as used by the hydrological model.

For model variants that applied the Penman-Monteith approach for estimating $E_{\text{ref}}$, relative humidity, global radiation and wind speeds were needed as further input data. Measured global radiation was used rather than global radiation derived from sunshine duration since for this study our interest is in the changes over time and, due to e.g. changes in the atmospheric aerosol concentrations over time (Norris and Wild, 2007), trends in sunshine duration may differ from those in global radiation. Measurements of relative humidity at 7:00 and 14:00 and global radiation were obtained from the ZAMG. Stations with more than 5 % (15 % for global radiation) missing data during 1976–2014 were excluded, which resulted in 125 and 6 stations for relative humidity and global radiation, respectively. Data gaps were filled using linear regression to the station with the highest correlation. The data were interpolated onto a 1 km$^2$ grid using local ordinary least-squares regression with elevation. The local neighbourhood was set to a default radius of 100 km for relative humidity and 200 km for global radiation, adjusted to include at

least 10 (global radiation 4) and at most 40 stations. Due to a strong influence of inhomogeneities, long-term changes in wind speed from measured wind speed data are highly uncertain (Böhm, 2008). This is also reflected in the fact that annual anomalies of wind speed data from 85 stations in Austria are hardly related to each other (Duethmann and Blöschl, 2018, see Supplement S1). Uniform monthly wind speeds averaged over all years from all stations in Austria were therefore applied in this study.

For an additional calibration to snow data, snow depth data from the HZB were interpolated by external drift kriging with elevation and aggregated to averages by elevation zone for each catchment (Parajka et al., 2007).

## 2.3    Hydrological model

### 2.3.1    Model description

In this study, we applied the same hydrological model as Merz2011, which is a semi-distributed conceptual model that follows the
structure of HBV (Hydrologiska Byråns Vattenbalansavdelning) (Bergström and Singh, 1995). The model equations can be found in Parajka et al. (2007). The model parameters are listed in Table 1. The model operates on a daily time step and the spatial discretization is based on 200 m elevation bands. Precipitation is partitioned into snow, rain or mixed precipitation based on air temperature using a lower and an upper threshold temperature $T_s$ and $T_r$. A snow correction factor SCF corrects undercatch of the precipitation gauges during snowfall. Snowmelt is calculated using a temperature-index approach based on the degree-day factor
DDF and the melt temperature $T_M$. Actual evaporation ($E_{sim}$) is estimated as a function of $E_{ref}$ and soil moisture. It equals $E_{ref}$ if soil moisture is above a calibrated threshold LP. Below this threshold, it linearly decreases to zero at a soil moisture level of zero. The fraction of the sum of rain and snowmelt that results in discharge is calculated as a nonlinear function of soil moisture. This involves the parameters FC, the maximum soil moisture storage, and the nonlinearity parameter $B$, where a larger $B$ is associated with a smaller fraction of direct runoff and vice versa. The runoff module consists of a hillslope component and a river routing
component. The hillslope component is represented by two linear stores that are connected through a constant percolation rate $C_p$. Fast runoff is generated if the state of the upper zone store is above a threshold LSUZ, using a fast storage coefficient $K_0$. Medium and slow runoff components are calculated as outflow from the upper and lower zone store, using the storage coefficients $K_1$ and $K_2$. In the river routing component, runoff routing in streams is simulated using a triangular transfer function involving the parameters $C_R$ and $B_{max}$.

### 2.3.2    Estimation of reference evaporation

Despite being technically external to the applied HBV model, the estimation of $E_{ref}$ is considered part of the hydrological model rather than part of the input data since it is calculated and not available as measured data. $E_{ref}$ is computed on a 1 km$^2$ grid and aggregated to elevation zones for each catchment, as used in the hydrological model. For the baseline model, $E_{ref}$ was derived based on a modified Blaney-Criddle method (DVWK, 1996), following Merz2011, denoted as E0

$$E0 = -1.55 + 0.96 \cdot (8.128 + 0.457 \cdot T) \cdot \frac{S_0 \cdot 100}{S_{\text{year}}} \tag{2}$$

where $T$ is the mean daily air temperature at 2 m height (°C), $S_0$ the potential daily sunshine duration (h), and $S_{\text{year}}$ is the mean yearly sum of potential sunshine duration (h).

In order to consider interannual variations in global radiation and vapour pressure deficit, in addition to air temperature, we calculated $E_{\text{ref}}$ using the Penman-Monteith equation for well-watered short grass vegetation (Allen et al., 1998), denoted as E1

$$E1 = 0.408 \cdot \frac{\Delta \cdot (R_n - G) + \gamma \cdot \frac{185400}{(T + 273) \cdot r_a} \cdot (e_s - e_a)}{\Delta + \gamma \cdot (1 + \frac{r_s}{r_a})} \tag{3}$$

where $R_n$ is the net radiation at the crop surface (MJ m$^{-2}$ d$^{-1}$), $G$ is the soil heat flux density (MJ m$^{-2}$ d$^{-1}$), $r_a$ is the aerodynamic resistance (s m$^{-1}$), $r_s$ is the surface resistance (s m$^{-1}$), $e_s$ is the saturation vapour pressure (kPa), $e_a$ is the actual vapour pressure (kPa), $\Delta$ is the slope of the vapour pressure curve (kPa °C$^{-1}$), and $\gamma$ is the psychrometric constant (kPa °C$^{-1}$). According to the reference conditions of a vegetated surface with a height of 0.12 m, $r_s = 70$ s m$^{-1}$ and $r_a = 208/u_2$ where $u_2$ is the wind speed at 2 m height (m s$^{-1}$), which was derived from the wind speed at 10 m height based on a logarithmic wind speed profile (Allen et al., 1998). The ground heat flux was neglected. The vapour pressure deficit $e_s - e_a$ was calculated as the average of the vapour pressure deficit at the minimum air temperature (using relative humidity at 7:00 LT) and at the maximum air temperature (using relative humidity at 14:00 LT). $R_n$ was estimated from global radiation ($R_s$; MJ m$^{-2}$ d$^{-1}$), albedo ($\alpha$; set to 0.23) and net longwave radiation ($R_{\text{nl}}$; MJ m$^{-2}$ d$^{-1}$)

$$R_n = (1 - \alpha) \cdot R_s + R_{\text{nl}} \tag{4}$$

where $R_{\text{nl}}$ was estimated according to Allen et al. (1998) based on minimum and maximum air temperature, clear-sky solar radiation, measured $R_s$, and the mean daily vapour pressure.

In order to consider additionally changes in the vegetation dynamics, we calculated $E_{\text{ref}}$ using a variable surface resistance based on changes in a satellite-based vegetation index (E2). We used observed 15-day maximum value composite data of the Normalized Difference Vegetation Index (NDVI) at a resolution of 8 km from the Advanced Very High Resolution Radiometer (AVHRR) from Tucker et al. (2005). For each point in time of this biweekly series, we aggregated the NDVI data to 200 m elevation zones based on the NDVI data for a rectangle around Austria. As the NDVI data are only available starting in 1981, we applied the data of July 1981–June 1982 for 1976–1981, where the NDVI data are not available. We used the parameterization from Sellers et al. (1996) to estimate a variable $r_s$ from the NDVI data. This involved estimating the fraction of photosynthetically active radiation (FPAR) from transformed NDVI data (Eq. (5); Sellers et al. (1996)), estimating the leaf area index (LAI) from the FPAR data (Eq. (6); Sellers et al. (1996)), and estimating $r_s$ from the LAI data (Eq. (7); Allen et al. (1998)).

$$\text{FPAR} = \frac{(S - S_{\min})}{(S_{\max} - S_{\min})} \cdot (\text{FPAR}_{\max} - \text{FPAR}_{\min}) + \text{FPAR}_{\min} \tag{5}$$

where $S$ is a transformed NDVI value $(1 + \text{NDVI})/(1 - \text{NDVI})$, and $S_{\min}$ and $S_{\max}$ are the 5 % and 98 % quantiles of $S$ for a given land cover class.

$$\text{LAI} = \text{LAI}_{\max} \cdot \frac{\log(1 - \text{FPAR})}{\log(1 - \text{FPAR}_{\max})} \tag{6}$$

where $\text{LAI}_{\max}$ is the maximum LAI of a land cover class. In Eq. (5) and Eq. (6), we applied the following coefficients for grassland: $\text{NDVI}_{\min} = 0.039$, $\text{NDVI}_{\max} = 0.674$, $\text{FPAR}_{\min} = 0.001$, $\text{FPAR}_{\max} = 0.95$, and $\text{LAI}_{\max} = 5$ (Sellers et al., 1996).

$$r_s = r_l \cdot (\text{LAI} \cdot 0.5)^{-1} \tag{7}$$

where $r_l$ is the leaf surface resistance. We applied a value of $r_l = 100$ s m$^{-1}$ for well-watered grass (Allen et al., 1998). Since the satellite based LAI values derived this way are often lower than the value of 2.88, which is assumed in the Penman-Monteith equation for well-watered short grass by Allen et al. (1998), E2 generally resulted in lower annual $E_{\text{ref}}$ than E0 or E1. In order to avoid water balance problems in the hydrological model, E2 was multiplied with the annual average ratio of E2 to E0 averaged over all catchments with a value of 1.2. Such an adjustment of $E_{\text{ref}}$ may be justified based on the fact that our study catchments are dominated by forest, and the maximum possible evaporation under well-watered conditions ($E_{\max}$) of forests is typically higher than $E_{\text{ref}}$ that assumes short grass. For example, analyses from non-weighable lysimeters suggest $E_{\max}$ to be 20 %–30 % higher for sites with pine forests at typical stand ages of 80–100 years compared to sites with grass (ATV-DVWK, 2001).

### 2.3.3 Model calibration

The objective function applied for model calibration consisted of three parts. An average of the Nash-Sutcliffe efficiency of linear and logarithmic discharge values ($f_Q$) was applied in order to achieve a balanced model performance for high and low flows. In order to keep the volume bias low, the absolute value of the relative volume bias ($f_{bias}$) was added as a penalty. Furthermore, a penalty for model parameters that deviate from an a priori distribution ($f_{beta}$) was added. The penalty function $f_{beta}$ is based on a Beta distribution for each parameter, as described in Merz2011. The a priori distributions for the model parameters were applied since, on the basis of the literature and previous applications of the model, we believe to have more information on the likely parameter values than just the parameter range. Including this criterion in the objective function has very little influence on the difference between simulated and observed discharge trends (Supplement S1). The objectives were combined in the following way

$$f_1 = w_1 \cdot (1 - f_Q) + w_2 \cdot f_{bias} + w_3 \cdot f_{beta} \tag{8}$$

setting the weights $w_1 = 0.8$, $w_2 = 1$, and $w_3 = 0.2$.

In order to test whether including annually aggregated discharge data in the objective function improves the model performance under transient climate conditions we additionally applied a modified objective function

$$f_2 = w_1 \cdot (1 - f_Q) + w_2 \cdot f_{bias} + w_3 \cdot f_{beta} + w_4 \cdot (1 - f_{annual}) \tag{9}$$

where $f_{annual}$ is the Nash-Sutcliffe efficiency calculated for discharge data aggregated to hydrological years. The weights were set to $w_1 = 0.4$, $w_2 = 1$, $w_3 = 0.1$, and $w_4 = 0.5$.

In a further model variant, we tested whether including snow data improves the model performance under transient climate conditions. The snow related part of the objective function aims at minimizing the number of days with poor snow cover simulations and was defined following Parajka et al. (2007). Observed snow cover was derived from maps of interpolated snow depth. An elevation zone was considered as snow covered if the average interpolated snow depth was greater than 0.5 mm, and snow free otherwise. In the model, an elevation zone was considered snow covered if the simulated snow water equivalent was

greater than 0.1 mm, and snow free otherwise. If the difference between simulated and observed snow cover on a particular day was greater than 50 % of the catchment area, it was considered as a day with poor snow cover simulations. The snow related part of the objective function $f_{snow}$ was defined as the ratio of the number of days with poor snow cover simulation and the number of days with observed snow cover. The overall objective function was then defined as

$$f_3 = w_1 \cdot (1 - f_Q) + w_2 \cdot f_{bias} + w_3 \cdot f_{beta} + w_4 \cdot f_{snow} \tag{10}$$

The weights were set to $w_1 = 0.7$, $w_2 = 1$, $w_3 = 0.1$, and $w_4 = 0.2$, following Parajka et al. (2007).

The objective function was minimized automatically with the shuffled complex evolution algorithm (SCE-UA) (Duan et al., 1992), a global optimization method based on the simplex downhill search scheme (Nelder and Mead, 1965). The calibration included 11 parameters. The upper and lower bounds and two further parameters of the Beta distribution for each parameter were selected following Merz2011 (Table 1). Four parameters that showed little sensitivity were pre-set to the following values: $T_R = 2°C$, $T_S = 0°C$, $C_r = 25$ $d^2$ $mm^{-1}$, and $B_{max} = 10$. As the focus of this study was on calibrating the model many times for different

calibration periods, catchments and model variants, characterizing parameter uncertainties was beyond the scope of this study. For the baseline model, we used seven consecutive 5-year calibration periods without temporal overlap (based on hydrological years), during 1978−2012. Each simulation was started with an additional 22-month warm-up period. As a modification, we also tested using a 25-year period as calibration period (1978−2002).

## 2.4 Analysing model problems for simulations under changing climate conditions

### 2.4.1 Metrics for evaluating model performance under changing climate conditions

Model performance was evaluated using the relative bias in discharge volume and the Nash-Sutcliffe efficiency (NSE). The relative bias in discharge volume was calculated as

$$bias = \left(\sum_{t=1}^{n} Q_{sim,t} - \sum_{t=1}^{n} Q_{obs,t}\right) / \sum_{t=1}^{n} Q_{obs,t} \tag{11}$$

where $Q_{sim,t}$ and $Q_{obs,t}$ are respectively the simulated and observed discharge on day $t$ and $n$ is the number of time steps.

In order to focus on the change in discharge under transient climate conditions, we used the difference between simulated and observed discharge trends as an additional criterion. Good performance in the calibration period but inability to estimate the changes in observed discharge resulting from the climatic changes indicates problems under transient climate conditions. Trends were evaluated over the entire study period (1978–2013). Trend significance was assessed by the nonparametric Mann-Kendall test (Mann, 1945; Kendall, 1975), and lag-one serial correlation was removed by applying the trend-free prewhitening technique (Yue et al., 2002). Trend slopes were estimated by the Sen's slope estimator (Sen, 1968). Uncertainties of the trend slope were estimated using a bootstrapping approach. For this purpose, 1000 samples of size $N$ were drawn, with replacement, from the record of length $N$ years and the Sen's slope was calculated for each of the 1000 samples. Then, the standard deviation was determined. Trends and the standard deviations were first derived for each catchment and then averaged over the catchments to determine average trends and their uncertainties over a number of catchments.

### 2.4.2 Hypotheses for the causes of the expected mismatch between observed and simulated discharge changes

We compiled possible explanations for the expected divergence between the observed and simulated changes in discharge based on the frameworks suggested by Westra et al. (2014) and Fowler et al. (2018) and the discussion in Coron et al. (2014). The working hypotheses are grouped into (1) data problems, (2) problems related to the model calibration, and (3) problems of the model structure (see Table 2). In a first analysis, the hypotheses were evaluated based on process understanding and literature. During this process, a number of the working hypotheses were rejected or assessed unlikely a cause of the differences between the observed and simulated discharge changes. Other hypotheses were evaluated using simulations with modifications of the model (Table 3).

(1) Data problems

Discharge data can be misleading if they are influenced by abstractions or streamflow diversions. For example, a general increase in water abstractions would reduce a positive streamflow trend. However, our study includes only catchments that were classified as devoid of substantial anthropogenic influences (Viglione et al., 2013) and any existing streamflow diversions were introduced

before the beginning of our study period (BMLFUW, 2015). Changes in water abstractions due to irrigation are not believed to be a major cause for the deviations between simulated and observed discharges as only about 3 % of the arable land in Austria is irrigated (FAO, 2016), the fraction of arable land is small in most of the study catchments (median 5 %, see Section 2.1) and the study catchments have only little overlap with those regions where irrigation is most relevant. These are small areas east, southeast and northwest of Vienna, where estimated average irrigation amounts of agricultural areas exceed 10 mm yr$^{-1}$ (BMLFUW, 2011). Erroneous trends in the discharge data could be caused by systematic trending errors of the rating curve. However, it seems unlikely that the discharge data of a large number of catchments are afflicted by systematic trends in the same direction. Problems in the discharge data were thus assumed unlikely to be a relevant cause for the differences between simulated and observed discharge trends.

Inhomogeneities of the precipitation data would result in biased estimates of the precipitation trends. A problem that would affect a large number of catchments is a varying number of precipitation stations included for generating the gridded precipitation data set. The precipitation data set used by Merz2011 was based on all available stations and included ~800 stations in the end of the 1970s and ~1050 stations around the year 2000 (Supplementary Figure S2). The effect of the changes in the number of stations on the trends in the water balance components was analysed by simulations with a precipitation data set based on all available stations (P0) and simulations with a precipitation data set based on a constant number of stations (P1). Changes in the gauge undercatch error due to changes in climate would also affect a large number of catchments. An increase of precipitation intensity and a decrease of the snow-to-rain ratio are expected to result in a higher catch ratio, meaning that the precipitation increase is lower than perceived by the observed data. The effect of neglecting the systematic precipitation error was estimated by simulations with a precipitation data set that is based on a constant number of stations that was corrected for the systematic gauge undercatch considering the influence of the precipitation type and daily precipitation intensity on the catch ratio (precipitation data set P2).

Similar to the precipitation data set, the air temperature data set in the baseline model was based on a variable station network, though the number of air temperature stations varies much less than the number of precipitation stations (Supplementary Figure S2). We investigated the effect of the changes in the number of air temperature stations by simulations with air temperature data sets based on all available stations (T0) or a constant number of stations (T1).

(2) Problems related to the model calibration

Problems in the model calibration relate to the problem that in principle parameter sets exist that allow good performance in the calibration and evaluation period but these parameter sets are not the ones identified during model calibration. Possible causes are, for example, a too short calibration period that results in overfitting, or processes that are relevant in the evaluation period but not activated in the calibration period. We therefore tested whether increasing the model calibration period from 5 yrs to 25 yrs reduces the bias between simulated and observed discharge trends. We furthermore investigated whether including annually

aggregated discharge data into the objective function improves the model performance under contrasting climate conditions, as found in a study by Hartmann and Bárdossy (2005). Since snow related processes are important in the mountainous part of the study area, we investigated further whether including data on interpolated snow depth into the objective function has an effect on the model performance under transient climate conditions. A recent study has shown that including snow data into the objective function can improve the temporal stability of snow related parameters (Sleziak et al., 2020).

(3) Problems of the model structure

In case the problem cannot be solved by rectifying problems in the data and model calibration, problems in the model structure are likely. These include inadequate process representations and changes in the catchment that are not represented by the model.

Differences between the observed and simulated trends in streamflow may result from a misconception of changes in $E_{\text{ref}}$. In Merz2011 as well as in the baseline model of our study, $E_{\text{ref}}$ is estimated using a modified Blaney-Criddle equation, which implies that interannual changes in $E_{\text{ref}}$ resulting from changes in other climate variables than air temperature are not accounted for. To consider effects of changes in global radiation and vapour pressure, we therefore additionally applied a more physically based method for estimating $E_{\text{ref}}$ using the Penman-Monteith equation (E1).

Further changes may result from changes in the vegetation dynamics as well as the land cover, such as a lengthening of the growing season, or increases in forest at the expense of cropland and extensive grassland, as observed in many parts of Austria (Krausmann et al., 2003; Gingrich et al., 2015). To test the possible effect of changes in vegetation dynamics on changes in the simulated trends of streamflow and evaporation, we performed additional simulations where we calculated a modified $E_{\text{ref}}$ considering changes in surface resistance based on a satellite-based vegetation index (E2). Land cover changes from agricultural land to forest may also contribute to changes in the satellite-based vegetation index. It is therefore assumed that the simulations with $E_{\text{ref}}$ considering changes in vegetation dynamics include also, to some extent, the effect of changes in land cover.

## 3 Results

### 3.1 Deviations between simulated and observed changes in discharge and evaporation of the baseline model

There is a clear gap between simulated and observed trends in discharge when the model calibrated in the first subperiod is applied to the entire period. On average over all catchments, the difference is $95 \pm 50$ mm yr$^{-1}$ per 35 yrs over 1978−2013 or $12.8 \pm 6.7$ % in relation to observed flow (Table 4). This is illustrated in Figure 2a that shows observed and simulated discharge for the model calibrated to 1978−1982 over the entire simulation period. Observed discharge of the 156 catchments showed only small increases over 1978–2013, with an average trend of $18 \pm 94$ mm yr$^{-1}$ per 35 yrs and significant ($p \leq 0.05$) increases and decreases in 10 % and 7 % of the catchments. In contrast, simulated discharge on average increased by $118 \pm 82$ mm yr$^{-1}$ per 35 yrs, with significant increases and decreases in 38 % and 1 % of the catchments. Discharge trends were overestimated by the model in

many catchments all over Austria (Figure 2c). Large differences between simulated and observed trends particularly occur in central Austria, southern Carinthia and western Tyrol.

The deviations in simulated and observed changes in discharge correspond to deviations in simulated and observed changes in evaporation. The dark blue line in Figure 2b shows the difference between precipitation and runoff, which may be interpreted as water-balance-based evaporation plus storage changes ($E_{wb}$). The fact that $E_{wb}$ includes storage changes and $E_{sim}$ does not, is relevant for short time scales but less so for long-term trends, as the fluctuations tend to average out over time. For example, the large interannual variations of $E_{wb}$ compared to $E_{sim}$ may be explained by storage changes. Large interannual variations are also observed for the difference between precipitation and simulated runoff, which is conceptually equivalent to $E_{wb}$ (Supplementary Figure S3). Comparing the long-term variations in $E_{wb}$ and $E_{sim}$, both $E_{wb}$ and $E_{sim}$ show increases, but $E_{sim}$ increased at a much lower rate than $E_{wb}$. Furthermore, the trend of $E_{wb}$ is reversed for the last two subperiods, whereas $E_{sim}$ increased over the entire simulation period. While the average trend of $E_{wb}$ over 1978–2013 is $139 \pm 59$ mm yr$^{-1}$ per 35 yrs, with significant increases in 76 % of the catchments, the average trend of $E_{sim}$ is $52 \pm 13$ mm yr$^{-1}$ per 35 yrs, with significant increases in 94 % of the catchments.

In order to investigate whether the overestimation of the simulated discharge trend is related to a decrease in simulated storage that is not represented by observed storage we examined simulated changes in storage. For this, we analysed the sum of all simulated storages, i.e. soil moisture storage, upper and lower zone subsurface storage and snow water equivalent, and calculated trends of annual averages (based on hydrological years). Trends in simulated storage were, on average over all catchments, $9 \pm 20$ mm over 1978–2013. This shows that the overestimation of the discharge trend is not generated by an opposite trend in simulate storage. Small changes in simulated storage are in agreement with no consistent large scale groundwater changes in the observations (Blaschke et al., 2011; Neunteufel et al., 2017).

While discharge volume biases during calibration were small, with average values over all catchments of 0.005–0.03 for the different subperiods, discharge biases during evaluation were much higher, with average values of −0.13–0.18 over the study catchments (Figure 3a). Curves of average bias during evaluation over the different subperiods for models calibrated in different subperiods show an interesting pattern. Average bias values during evaluation increase from subperiod S1 to S6 by 0.15–0.18 and decrease again for the last period. The curves run almost parallel and differ by a vertical offset that ensures low bias during the calibration period. The changes in the average bias were not caused by few catchments with very large changes, as shown by changes in the distribution of bias across all catchments (Supplementary Figure S4). NSE values during model calibration varied in the range of 0.70–0.75 on average over the catchments, showing that the model performed well in each subperiod when calibrated to it. As expected, model performance during evaluation was lower, with average values over the study catchments of 0.56–0.71 (Figure 3b). In many cases, model performance decreases with increasing distance between the calibration and the evaluation period, particularly for model evaluations in subperiods S1 and S2.

The performance of the baseline model agrees well with the study by Merz2011, who found average NSE during model calibration of 0.74–0.77 and average NSE during model evaluation of 0.64–0.69, when evaluating over all subperiods except the one used for calibration (compared to 0.70–0.75 during calibration and 0.63–0.66 during evaluation in our study). Discharge biases during calibration were slightly smaller in the present study, due to including a penalty for discharge bias in the objective function. The longer study period used in our study revealed that the trend of an increasing difference between simulated and observed discharge, when applying the model calibrated in subperiod S1 to the entire study period, was not continued during the last subperiod.

## 3.2    Data problems

### 3.2.1    Precipitation

Driving the hydrological model with a precipitation data set based on a variable number of precipitation stations may influence the estimated trend of precipitation and thus the trend of simulated discharge. In order to quantify this effect, we performed model simulations with a precipitation data set based on a constant number of stations (P1) in comparison to the baseline precipitation data set P0 that uses a variable number of stations. This reduced the gap between simulated and observed discharge from $95 \pm 50$ mm yr$^{-1}$ per 35 yrs to $55 \pm 47$ mm yr$^{-1}$ per 35 yrs (Table 4), i.e. a reduction by $39 \pm 26$ mm yr$^{-1}$ per 35 yrs (Table 5). The reduced gap between simulated and observed discharge is consistent with the difference in the trends in the precipitation data sets. The baseline precipitation data set P0 suggests a precipitation increase of on average $159 \pm 89$ mm yr$^{-1}$ per 35 yrs, whereas the precipitation data set P1 results in an increase of $121 \pm 89$ mm yr$^{-1}$ per 35 yrs (Figure 4a). Better model performance with respect to changes in streamflow volume is also reflected by smaller increases in bias during evaluation in the different subperiods (Figure 5a).

Changes in the snow-to-rain ratio and in the precipitation intensity may affect the undercatch error and thus the precipitation trend. Figure 4c−e shows that, over the study period, the snow-to-rain ratio decreased and the daily precipitation intensity increased, whereas the number of precipitation days remained relatively stable. In the precipitation data sets P0 and P1, the precipitation undercatch error is neglected. In order to estimate the magnitude of the effect of changes in air temperature and precipitation intensity on changes of the undercatch error, we performed simulations with a precipitation data set that was corrected for undercatch accounting for daily precipitation intensity and precipitation type, which was estimated based on air temperature (precipitation data set P2). Precipitation data set P2 exhibits generally higher precipitation and, with an average trend of $120 \pm 93$ mm yr$^{-1}$ per 35 yrs, a similar absolute and a lower relative precipitation increase over time compared to the precipitation data set P1 (Figure 4a). Simulations with precipitation data set P2 resulted in a gap between simulated and observed discharge trends of $48 \pm 47$ mm yr$^{-1}$ per 35 yrs (Table 4), i.e. a reduction by $47 \pm 28$ mm yr$^{-1}$ per 35 yrs compared to the baseline model V0 that uses precipitation data set P0 (Table 5). Comparing model variants V2 to V0, strong reductions of the differences between simulated

and observed discharge trends particularly occurred in catchments where the differences between simulated and observed discharge trends were large (Supplementary Figure S6d, Figure 2c). The tendency to further reduce the gap compared to simulations with the precipitation data set P1 of $8 \pm 9$ mm yr$^{-1}$ per 35 yrs was not significant.

### 3.2.2 Air temperature

In order to investigate the possible effect of changes in the station network for air temperature data, we performed simulations with gridded air temperature data based on stations with a complete record over the study period (T1), as compared to simulations with a gridded data set based on all available air temperature series (T0). This showed virtually no differences in discharge trends between the two variants (Table 4). The small effect of varying the air temperature data set can be explained by the fact that changes in the station network were only small (Supplementary Figure S2) and the two data sets result in very similar changes over time (Figure 4b).

## 3.3 Problems of the model calibration

### 3.3.1 Varying the length of the calibration period

In order to evaluate whether the calibration period was too short, we increased the calibration period from 5 yrs (1978–1982) to 25 yrs (1978–2002) (model variant V4). This resulted in an average discharge trend of $113 \pm 82$ mm yr$^{-1}$ per 35 yrs over 1978–2013 (Table 4) and thus virtually no effect compared to the baseline model.

### 3.3.2 Varying the objective function

Changing the objective function by including annually aggregated discharge data (model variant V5) led to an average discharge trend of $115 \pm 83$ mm yr$^{-1}$ per 35 yrs over 1978–2013 (Table 4) and thus no improvement in the simulation of the long-term discharge trends either.

Including a snow related criterion into the objective function (model variant V6) improved the model performance with respect to snow without deteriorating the model performance for discharge (Supplementary Table S1). The performance of the model compared to observed snow cover derived from interpolated snow depth was comparable to Parajka et al. (2007), when considering the same set of catchments. Model performance with respect to long-term trends was not improved, with an average gap between simulated and observed discharge trends of $95 \pm 50$ mm yr$^{-1}$ per 35 yrs over 1978–2013 (Table 4).

### 3.4 Problems of the model structure

#### 3.4.1 Calculation of $E_{ref}$ using the Penman-Monteith equation

To estimate the effect of using a simplified versus a more physically-based equation for estimating $E_{ref}$, we compared simulations with $E_{ref}$ estimated by the Blaney-Criddle method (simulation V0) to simulations with $E_{ref}$ estimated by the Penman-Monteith method (model variant V7). The results showed only negligible differences between the two model variants in terms of simulated discharge trends (Table 4). This is consistent with small differences between the trends in $E_{ref}$ estimated by the two different methods, with average trends of $70 \pm 13$ mm yr$^{-1}$ per 35 yrs for E0 (Blaney-Criddle) and $71 \pm 17$ mm yr$^{-1}$ per 35 yrs for E1 (Penman-Monteith) (Figure 6).

#### 3.4.2 Calculation of $E_{ref}$ considering changes in vegetation dynamics

In order to consider changes in the vegetation dynamics, we estimated changes in surface resistance based on changes in a satellite-based vegetation index for the calculation of $E_{ref}$. Accounting for vegetation dynamics in the calculation of $E_{ref}$ increased trends in $E_{sim}$ to $88 \pm 16$ mm yr$^{-1}$ per 35 yrs (model variant V8), compared to $52 \pm 13$ mm yr$^{-1}$ per 35 yrs in the baseline model V0 (Table 4). This reduced the gap between simulated and observed discharge trends from $95 \pm 50$ mm yr$^{-1}$ per 35 yrs to $58 \pm 49$ mm yr$^{-1}$ per 35 yrs (Table 4), i.e. a reduction by $36 \pm 9$ mm yr$^{-1}$. Increased trends in $E_{sim}$ are consistent with $E_{ref}$ trends that increased from $70 \pm 13$ mm yr$^{-1}$ per 35 yrs in the baseline model V0 to $110 \pm 17$ mm yr$^{-1}$ per 35 yrs in model variant V8 (Figure 6). Accounting for vegetation dynamics had a rather consistent effect on the discharge trends throughout the catchments (Supplementary Figure S6b and e). In order to evaluate the effect of combining the model modifications that had a considerable effect on the gap between trends in observed and simulated discharge, we combined the use of the precipitation data set P2 (model variant V2) and the consideration of vegetation dynamics in the calculation of $E_{ref}$ (model variant V8) as model variant V9. Compared to the baseline model, the differences in trends between simulated and observed discharge were reduced by $90 \pm 31$ mm yr$^{-1}$ per 35 yrs in this model variant so that the differences largely disappeared (Table 4). Bias values in the evaluation period for variant V9 show only little variation between subperiod S2 to S6, but some variation remains when transferring models from subperiods S1 or S7 to subperiod S2 to S6, or vice versa (Figure 5h). Bias values in the evaluation period were reduced from $-0.13$–$0.18$ in the baseline model to $-0.03$–$0.10$ in model variant V9. Comparing model variant V9 and the baseline V0, the differences in trends of simulated and observed discharge were reduced in most catchments, with stronger reductions in catchments that showed higher differences in trends of simulated and observed discharge in the baseline model (Supplementary Figure S6f).

# 4 Discussion

Our analyses suggest that problems in the precipitation data and neglecting changes in vegetation activity were the most important causes of the poor performance of the HBV model in Austrian catchments in a transient climate. Inhomogeneities in the precipitation data set due to a variable number of stations explained $39 \pm 26$ mm yr$^{-1}$ per 35 yrs of the difference between simulated and observed discharge trends (or $47 \pm 28$ mm yr$^{-1}$ per 35 yrs when using a precipitation data set that was additionally undercatch corrected). While the original model neglected changes in the vegetation activity and length of the growing season, considering these changes by calculating $E_{ref}$ accounting for changes in surface resistance based on changes in a satellite-based vegetation index reduced the gap between simulated and observed discharge trends by $36 \pm 9$ mm yr$^{-1}$ per 35 yrs. Combining both modifications, using a precipitation data set based on a constant number of stations and considering vegetation dynamics for the calculation of $E_{ref}$, reduced the gap between simulated and observed discharge trends by 95 %.

The model structure deficiencies with respect to vegetation dynamics are likely relevant for a large number of studies in a transient climate, including simulations in the context of climate change impact assessments. In a changing climate, changes in vegetation dynamics (such as increased growing season length) can have substantial effects on changes in the water balance. The effect of considering changes in vegetation dynamics observed in this study is in agreement with other studies that demonstrate impacts of climate-induced changes in growing season length and vegetation growth on the water balance (Caldwell et al., 2016; Hwang et al., 2018; Kim et al., 2018; Gaertner et al., 2019). For example, long-term hydrologic changes in two forested catchments in the southern Appalachians could only be simulated if full vegetation dynamics were incorporated in the eco-hydrologic model (Hwang et al., 2018). Lengthening of the growing season intensified climatically driven increases in evaporation and reductions in streamflow in a mixed forest catchment in New England (Kim et al., 2018). Decreased catchment streamflow over the last 15 years was linked to increased growing season length in six northern headwater catchments (Wang et al., 2019). Increases in evaporation in the central Appalachian Mountain region were attributed to longer growing seasons, with an increase of growing season length of 1 day resulting in a moderate increase of evaporation of 0.5 mm yr$^{-1}$ (Gaertner et al., 2019). Here, we considered changes in vegetation dynamics by using a variable surface resistance based on changes in a satellite-based vegetation index. Based on a rather simple approach, this should be seen as a first estimate to demonstrate the significance of changes in vegetation dynamics on the water balance. While in this study we assume that the simulations accounting for vegetation dynamics also partly reflect the effects of changes in land cover, an approach that allows disentangling these effects would be preferable in future work. The changes in vegetation dynamics were derived from satellite-based data, which are often not available in the context of climate change impact assessments. Future work should therefore aim at approaches that simulate the changes in vegetation dynamics in response to climatic changes that may be implemented into conceptual hydrologic models. The effect of increased atmospheric $CO_2$ concentrations on surface resistance was neglected in the present study. At the global scale, it is estimated that this effect may have reduced evaporation in the order of 1.6 to 2.0 mm yr$^{-1}$ decade$^{-1}$ since the 1960s (Gedney et al., 2006; Piao et al., 2007).

In this study, we found problems in the model structure with respect to the calculation of evaporation to contribute to poor model performance in a transient climate. Model structural problems albeit in different model components were also found to cause poor performance in a transient climate in other studies. For a case study in South Australia, model performance was improved by allowing the parameter for the maximum capacity of the soil store to vary in time as a function of a linear trend, which was interpreted as increased catchment storage through an increase in farm dams in the catchment (Westra et al., 2014). For a case study in southwest Australia, introducing a nonlinearity parameter and a threshold value for the rainfall-runoff relationship enabled the simulation of dry and non-dry years with the same parameter set, which was not possible with the original model (Fowler et al., 2018). Changes in glacier volume may cause deviations between simulated and observed discharge trends if not accounted for by the model. Therefore, glacier covered catchments were excluded in our study. Model structural deficits with respect to glacier dynamics may be responsible for further deviations between simulated and observed discharge trends in the study by Merz2011, which did not exclude glacier covered catchments, although the total glacier cover of Austria is small (0.5 %; Fischer et al. (2015)).

The mismatch between simulated and observed discharge trends was partly caused by inhomogeneities in the precipitation data. Thus, the problem of the limited suitability of the hydrological model under transient conditions is less severe than previously assumed. The comparison of the precipitation data sets based on a constant and variable station network (Figure 4a) shows very well that trend analyses of gridded data based on a variable number of stations can be misleading. Particularly large effects of changes in the gauge network on estimated trends may occur if the gauged precipitation values are interpolated directly (as for the baseline precipitation data P0), in contrast to interpolation methods that make use of a two-step procedure by interpolating against a climatology (Fawcett et al., 2010). While the SPARTACUS data are currently seen as the best-suited gridded data set for trend analyses in Austria, they may however contain further inhomogeneities. Network inhomogeneities were avoided by using a constant station network and interpolating against a monthly climatology. However, inhomogeneities may be present in the series of individual stations. Homogenized series were available only for 4 % of the station data used for the SPARTACUS data set, and it is estimated that 25 % of the stations used may still be affected by inhomogeneities (Hiebl and Frei, 2018). However, while we expect changes in the precipitation trends for individual (smaller) catchments, it seems unlikely that inhomogeneities in the station data cause changes in the precipitation trends in the same direction for a large number of catchments.

Considering the precipitation undercatch error including effects of climate variability on the undercatch error had a small and not significant effect, when compared to the simulation using the same precipitation data without undercatch correction. Since high quality wind speed data were not available, wind speeds were not considered in the calculation of the undercatch error. Analyses of the available data in Austria over 1977–2014 show a slight decrease in wind speeds (on average -3.0 ± 2.5 % per decade, see Supplement S2 in Duethmann and Blöschl (2018)). Decreasing wind speeds would result in increasing catch ratios and mean that our estimate of the effect of changes in the catch ratio due to climatic variability on the difference between simulated and observed discharge trends is at the lower end.

Increasing the length of the calibration period did not reduce the gap between trends in simulated and observed discharge (Table 4). This is in agreement with several other studies that found little improvement of the observed poor performance in contrasting climate by using a longer calibration period (Luo et al., 2012; Brigode et al., 2013; Coron et al., 2014). Similarly, changes to the objective function to improve the internal consistency of the model did not lead to a better performance in a changing climate. In this study, we included snow data because of the influence of snow on the hydrology in the study region. Seibert (2003) tested whether including groundwater-level observations in the calibration reduced their problem of low model performance for large floods, when there were no large floods in the calibration period, but this did not lead to improvements. The results are more variable with respect to changes in the objective function that put a stronger focus on interannual variability. While including annually aggregated discharge data into the objective function did not reduce the gap between trends in simulated and observed discharge in this study, Hartmann and Bárdossy (2005) found that including annually aggregated discharge data in the objective function in addition to daily discharge data improved the transferability of a distributed conceptual hydrological model under contrasting climate conditions in their study. A way to find out whether parameter problems might be the cause when a model shows poor performance in contrasting climates is to apply multiobjective calibration to the contrasting periods, as suggested by Fowler et al. (2018). If this is the case, efforts of finding a parameterization method that identifies parameter sets suitable for contrasting climates only from the calibration period may then be undertaken in a second step. Multiobjective calibration to the contrasting periods was applied in a study that used five different model structures and 86 catchments in Australia (Fowler et al., 2016). The results showed that depending on the acceptance threshold for good model performance, parameterization problems caused a decline in model performance in contrasting climate periods in 35 % or 55 % of the cases of DSST failure.

The present study included a large number of catchments, so we assume that our results are robust. However, it is limited to a particular hydrologic model and a particular region. It should therefore be complemented by further studies on the causes of poor (and good) performance of hydrological models in transient climate conditions. The aim is a more complete picture on in what cases what model structure components and what parameterization methods result in poor model performance in a transient climate so that these model structure components and parameterization methods can be avoided for applications where good model performance in a transient climate is relevant, as for example in climate change impact assessments. Ultimately, this will increase the robustness of hydrologic simulations in a changing climate.

## 5    Conclusion

In this study, we investigated why the HBV model failed to predict changes in discharge in response to observed increases in precipitation and air temperature for 156 catchments in Austria. The baseline model overestimated the observed discharge trends over 1978–2013 and on average over all catchments by $95 \pm 50$ mm yr$^{-1}$ per 35 yrs, or $12.8 \pm 6.7$ % per 35 yrs relative to observed discharge. Simulations with variants of the model indicate that the poor performance of the HBV model in Austrian catchments in a transient climate could largely be ascribed to two problems, a model structure that neglects changes in the vegetation dynamics,

and inhomogeneities in the precipitation input. Considering changes in the vegetation dynamics by calculating $E_{ref}$ accounting for changes in surface resistance based on changes in a satellite-based vegetation index reduced the gap between simulated and observed discharge trends by $36 \pm 9$ mm yr$^{-1}$ per 35 yrs. Inhomogeneities in the precipitation data set due to a variable number of stations on average explained $39 \pm 26$ mm yr$^{-1}$ per 35 yrs of the difference between simulated and observed discharge trends. Extending the calibration period from 5 to 25 yrs, including annually aggregated discharge data or snow cover in the objective function, or estimating evaporation with the Penman-Monteith instead of the Blaney-Criddle approach had little influence on the simulated discharge trends. The model structure deficiencies with respect to vegetation dynamics are likely relevant for a large number of studies in a transient climate, including climate change impact studies. The precipitation data problem highlights the importance of using precipitation data based on a constant number of stations for studies on long-term dynamics. Our study emphasizes the importance of considering interrelations between changes in climate, vegetation and hydrology for hydrological modelling in a transient climate.

**Data availability**. The discharge data and precipitation data from the HZB can be accessed through https://ehyd.gv.at/ (last access: 31 May 2020). The meteorological data from the ZAMG are currently not freely available; requests should be directed to klima@zamg.ac.at. The Corine land cover map can be downloaded from https://www.eea.europa.eu/data-and-maps/data/clc-2000-vector-6 (last access: 31 May 2020). The SRTM DEM can be obtained from http://srtm.csi. cgiar.org (last access: 31 May 2020). The NDVI data can be downloaded from https://ecocast.arc.nasa.gov/data/pub/gimms/. The hydrological model simulations are available upon request from the first author.

**Author contributions.** DD conceived and designed the study, performed the analyses, and prepared the manuscript. GB contributed to the study design and interpretation of the results. JP contributed to the numerical analyses. All authors actively took part in the discussion of the results and revising the paper.

**Competing interests.** The authors declare that they have no conflict of interest.

**Acknowledgements.** We thank Mojca Sraj, David Post, Chang Liao, Yan Liu, Veit Blauhut, Amelie Herzog, Tunde Olarinoye Ruth Stephan, Taehee Hwang, an anonymous referee and the editor Matjaž Mikoš for their comments that helped to improve the manuscript. We gratefully acknowledge the financial support from the DFG (German Research Foundation) through a research scholarship to DD (DU 1595/1-1). We would like to thank the Central Hydrographical Bureau and the Austrian Central Institute for Meteorology and Geodynamics for providing the hydrographic and meteorological data.

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

# Tables

**Table 1** A priori distribution of parameter values where $p_l$ and $p_u$ are the lower and upper bounds, $\alpha$ and $\beta$ the parameters of the a priori distribution, and $p_{max}$ the parameter value at which the a priori distribution is at its maximum. Note that the parameters $T_R$, $T_S$, $C_r$ and $B_{max}$ were set constant and are therefore not listed here.

| Parameter | Unit | Description | $p_l$ | $p_u$ | $p_{max}$ | $\alpha$ | $\beta$ |
|---|---|---|---|---|---|---|---|
| SCF | - | Snow correction factor | 1 | 1.5 | 1.03 | 1.1 | 2.5 |
| DDF | mm (°C d)$^{-1}$ | Degree-day factor | 0.5 | 5 | 1.25 | 1.5 | 3.5 |
| $T_m$ | °C | Melt temperature | -2 | 2 | 0 | 2 | 2 |
| FC | mm | Maximum soil moisture storage | 0 | 600 | 150 | 1.05 | 1.15 |
| LP/FC | - | Ratio of limit for $E_{ref}$ and FC | 0 | 1 | 0.94 | 4 | 1.2 |
| $B$ | - | Nonlinearity parameter of runoff generation | 0 | 20 | 3.4 | 1.1 | 1.5 |
| $K_0$ | days | Very fast storage coefficient of additional outlet | 0 | 2 | 0.5 | 2 | 4 |
| $K_1$ | days | Fast storage coefficient | 2 | 30 | 9 | 2 | 4 |
| $K_2$ | days | Slow storage coefficient | 30 | 250 | 105 | 1.05 | 1.05 |
| $C_p$ | mm d$^{-1}$ | Percolation rate | 0 | 8 | 2 | 2 | 4 |
| LSUZ | mm | Storage capacity threshold | 1 | 100 | 50 | 3 | 3 |

**Table 2** Working hypotheses for potential causes of the divergence between observed and simulated discharge changes.

| Working hypothesis | Analysis or further explanation |
| --- | --- |
| **(1) Data problems** | **→ Section 3.2** |
| **(1.1) Problems in the discharge data** | |
| Changes in abstractions or diversions | Catchments with anthropogenic influences were generally excluded. Reviewed comments in the hydrological yearbooks: diversions were introduced before the start of the study period. Only a small fraction of the arable land in Austria is irrigated and this does largely not overlap with the study catchments |
| Rating curve errors | Rating curve errors unlikely to occur in the same direction for a large number of catchments. → Unlikely to be relevant for a large number of catchments. |
| **(1.2) Problems in the precipitation data** | |
| Inhomogeneities in the precipitation data due to instrument changes | Introduction of heated precipitation gauges → Would result in larger precipitation increases and thus increase the gap between changes in $E_{wb}$ and changes in $E_{sim}$. Since at most locations with a heated gauge, there is a manually operated gauge in addition and values of the latter are used to report daily precipitation sums, this effect is likely not relevant. |
| Inhomogeneities in the gridded precipitation data due to changes in the number of stations | Simulations with a precipitation data set that uses a constant number of stations (model variant V1) |
| Biased estimates of the precipitation trend due to changes in the catch ratio caused by changes in the snow-to-rain ratio and changes in precipitation intensities (in addition to inhomogeneities due to a variable number of stations) | Simulations with a precipitation data set with a constant number of stations and correction for the systematic precipitation undercatch (considering the precipitation type and precipitation intensity (based on daily precipitation amount)) (model variant V2) |
| **(1.3) Problems in the air temperature data** | |
| Inhomogeneities in the gridded air temperature data due to changes in the number of stations | Simulations with a data set that uses a constant number of stations (model variant V3) |
| **(2) Problems related to the model calibration** | **→ Section 3.3** |
| Too short calibration period | Simulations with a 25-year calibration period (model variant V4) |
| Objective function insensitive to long-term discharge variations | Simulations with a modified objective function that includes annually aggregated discharge data (model variant V5) |
| Internal inconsistencies due to calibration only to discharge | Simulations with a modified objective function that includes a comparison against snow data (model variant V6) |
| **(3) Problems of the model structure** | **→ Section 3.4** |
| Effects of changes in radiation and saturation deficit not reflected by the model | Calculation of $E_{ref}$ with the Penman-Monteith approach (model variant V7) |
| Effects of changes in the vegetation dynamics and land cover not reflected by the model | Calculation of $E_{ref}$ using a variable surface resistance based on a satellite-derived vegetation index (model variant V8) |

**Table 3** Overview of model variants.

| Abbreviation | Description | Input precipitation | Input air temperature | Length of calibration periods | Objective function | Calculation of $E_{ref}$ |
|---|---|---|---|---|---|---|
| V0 | Baseline model | P0 | T0 | 5 yrs | $f_1$ | E0 |
| V1 | Vary $P$ data set | P1 | T0 | 5 yrs | $f_1$ | E0 |
| V2 | Include $P$ undercatch correction | P2 | T0 | 5 yrs | $f_1$ | E0 |
| V3 | Vary air temperature data | P0 | T1 | 5 yrs | $f_1$ | E0 |
| V4 | Increase length of calibration period | P0 | T0 | 25 yrs | $f_1$ | E0 |
| V5 | Include annually aggregated $Q$ into obj. function | P0 | T0 | 5 yrs | $f_2$ | E0 |
| V6 | Include snow into obj. function | P0 | T0 | 5 yrs | $f_3$ | E0 |
| V7 | $E_{ref}$ based on Penman-Monteith | P0 | T0 | 5 yrs | $f_1$ | E1 |
| V8 | Modified $E_{ref}$ dependent on NDVI | P0 | T0 | 5 yrs | $f_1$ | E2 |
| V9 | Combine V2 and V8 | P2 | T0 | 5 yrs | $f_1$ | E2 |

**Table 4** Linear trends in water balance components (mm yr$^{-1}$ per 35 yrs) over 1978–2013 as averages over all catchments. Simulated values refer to the model calibrated in subperiod S1 1978–1982. Uncertainties relate to standard deviations of the trend slope averaged over all catchments. For trends in $Q_{sim} - Q_{obs}$, we first derived series of the differences $Q_{sim} - Q_{obs}$ for each catchment and then estimated trends.

| | $P_{obs}$ | $E_{ref}$ | $Q_{obs}$ | $E_{wb}$ | $Q_{sim}$ | $E_{sim}$ | $Q_{sim} - Q_{obs}$ |
|---|---|---|---|---|---|---|---|
| V0 Baseline model | 159 ± 89 | 70 ± 13 | 18 ± 94 | 139 ± 59 | 118 ± 82 | 52 ± 13 | 95 ± 50 |
| V1 Vary *P* data set | 121 ± 89 | 70 ± 13 | 18 ± 94 | 97 ± 57 | 80 ± 81 | 51 ± 14 | 55 ± 47 |
| V2 Include *P* undercatch correction | 120 ± 93 | 70 ± 13 | 18 ± 94 | 96 ± 57 | 72 ± 85 | 59 ± 13 | 48 ± 47 |
| V3 Vary air temperature data | 159 ± 89 | 70 ± 13 | 18 ± 94 | 139 ± 59 | 117 ± 82 | 54 ± 13 | 93 ± 50 |
| V4 Increase length of calibration period | 159 ± 89 | 70 ± 13 | 18 ± 94 | 139 ± 59 | 113 ± 82 | 59 ± 14 | 89 ± 51 |
| V5 Include annually aggregated *Q* into obj. function | 159 ± 89 | 70 ± 13 | 18 ± 94 | 139 ± 59 | 115 ± 83 | 54 ± 14 | 93 ± 49 |
| V6 Include snow into obj. function | 159 ± 89 | 70 ± 13 | 18 ± 94 | 139 ± 59 | 118 ± 82 | 53 ± 14 | 95 ± 50 |
| V7 $E_{ref}$ based on Penman-Monteith | 159 ± 89 | 71 ± 17 | 18 ± 94 | 139 ± 59 | 113 ± 84 | 53 ± 14 | 92 ± 49 |
| V8 Modified $E_{ref}$ dependent on NDVI | 159 ± 89 | 110 ± 17 | 18 ± 94 | 139 ± 59 | 80 ± 83 | 88 ± 16 | 58 ± 49 |
| V9 combine V2 and V8 | 120 ± 93 | 110 ± 17 | 18 ± 94 | 96 ± 57 | 26 ± 86 | 104 ± 17 | 4 ± 46 |

**Table 5** Working hypotheses for potential causes of the divergence between observed and simulated discharge changes that were further analysed and estimated magnitude of the effect on the gap between trends in $Q_{obs}$ and $Q_{sim}$ (mm yr$^{-1}$ per 35 yrs) over 1978–2013 compared to the baseline model. This was calculated by deriving series of the differences in annual discharge of the respective model variant compared to the baseline model (e.g., $Q_{sim,V1} - Q_{sim,V0}$) for each catchment and then estimating trends. Uncertainties relate to standard deviations of the trend slope averaged over all catchments.

| Working hypothesis | Model variant | Result | Magnitude of the effect (mm yr$^{-1}$ per 35 yrs) |
|---|---|---|---|
| **(1) Data problems** | | → Section 3.2 | |
| **(1.2) Problems in the precipitation data** | | | |
| Inhomogeneities in the gridded precipitation data due to changes in the number of stations | V1 | Reduces the gap between changes in $Q_{obs}$ and $Q_{sim}$ | ↓−39 ± 26 |
| Biased estimates of the precipitation trend due to changes in the catch ratio caused by changes in the snow-to-rain ratio and changes in precipitation intensities (in addition to inhomogeneities due to a variable number of stations) | V2 | Reduces the gap between changes in $Q_{obs}$ and $Q_{sim}$ | ↓ −47 ± 28 |
| **(1.3) Problems in the air temperature data** | | | |
| Inhomogeneities in the gridded air temperature data due to changes in the number of stations | V3 | Little effect on simulated discharge trends | −1 ± 5 |
| **(2) Problems related to the model calibration** | | → Section 3.3 | |
| Too short calibration period | V4 | Little effect on simulated discharge trends | −5 ± 9 |
| Objective function insensitive to long-term discharge variations | V5 | Little effect on simulated discharge trends | −3 ± 13 |
| Internal inconsistencies due to calibration only to discharge | V6 | Little effect on simulated discharge trends | 0 ± 4 |
| **(3) Problems of the model structure** | | → Section 3.4 | |
| Effects of changes in radiation and saturation deficit not reflected by the model | V7 | Little effect on simulated discharge trends | −2 ± 7 |
| Effects of changes in the vegetation dynamics and land cover not reflected by the model | V8 | Reduces the gap between changes in $Q_{obs}$ and $Q_{sim.}$ | ↓ −36 ± 9 |

**Figures**

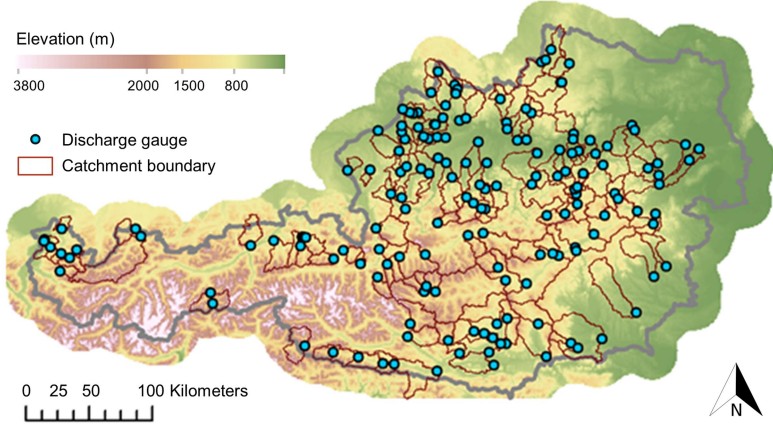

**Figure 1** Distribution of the study catchments in Austria.

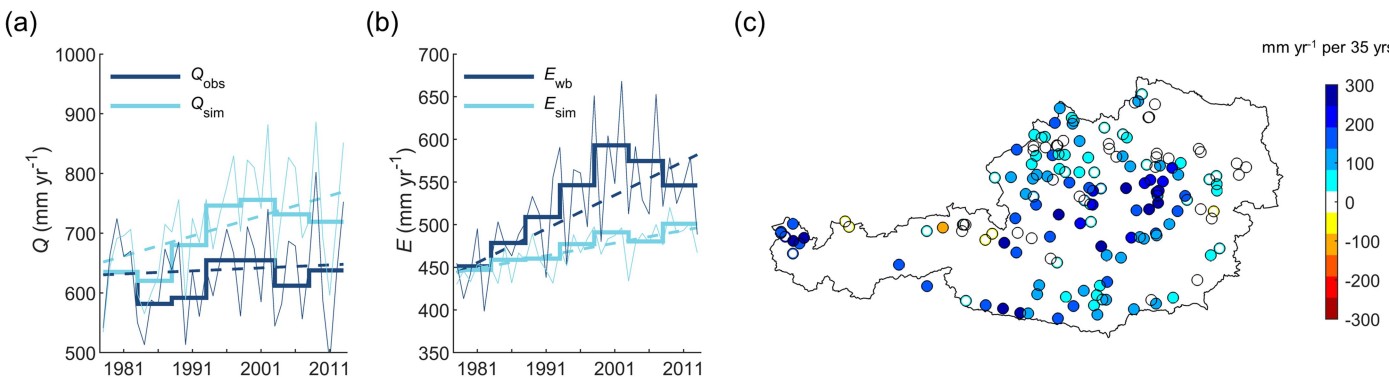

**Figure 2** (a) Temporal variations in simulated discharge ($Q_{sim}$) and observed discharge ($Q_{obs}$), as averages over all 156 study catchments. (b) Temporal variations in simulated evaporation ($E_{sim}$) and evaporation derived from the water balance ($E_{wb}$), as averages over all study catchments. Note that $E_{wb}$ includes storage changes that are particularly relevant for the interannual variations. The thick lines show subperiod annual means, the thin lines annual sums, and the dashed lines linear trends. (c) Spatial pattern of the differences of simulated and observed trends in discharge. Filled circles indicate significant trends at $p \leq 0.05$.

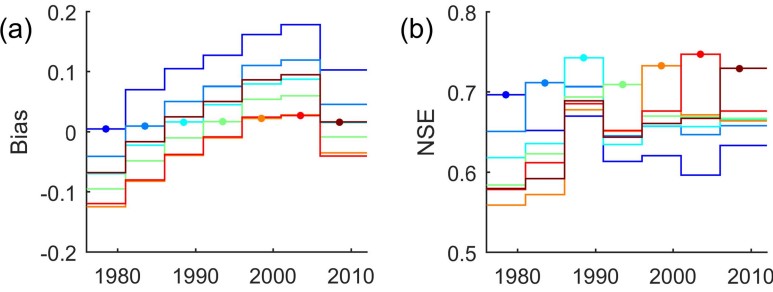

**Figure 3** (a) Bias and (b) NSE for the different subperiods averaged over all study catchments for the baseline model V0. Each line refers to models calibrated in one subperiod, showing bias and NSE during calibration (marked by the filled circle) and during evaluation in the other six subperiods.

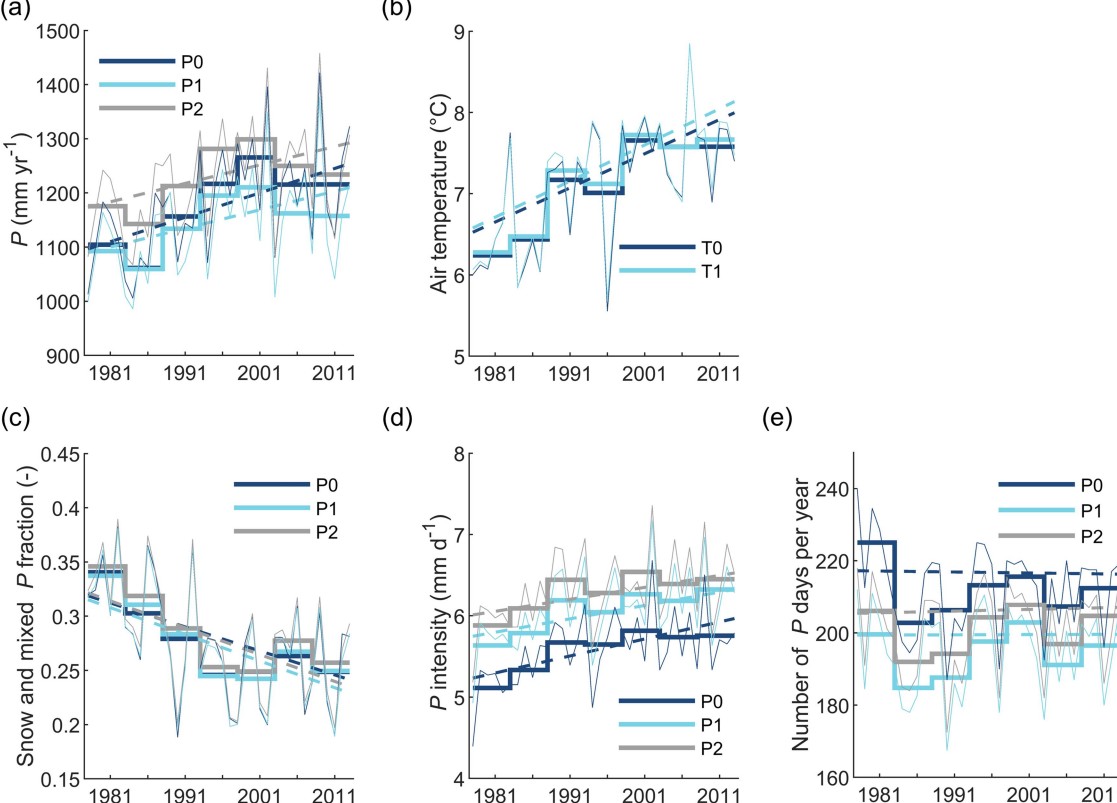

**Figure 4** Temporal variations of (a) precipitation, (b) air temperature, (c) fraction of snow and mixed precipitation (estimated as precipitation on days with average daily air temperatures below 3°C), (d) precipitation intensity (precipitation day defined as day with precipitation ≥ 0.1 mm $d^{-1}$), (e) number of precipitation days per year; as represented by different data sets, averaged over all catchments. The thick lines show subperiod means, the thin lines annual sums, and the dashed lines linear trends, the different colours represent different data sets. Precipitation data set P0 is based on a variable number of stations over time, P1 is based on a constant number of stations, and P2 is based on a constant number of stations and includes a correction for undercatch. Air temperature data set T0 is based on a variable number of stations and T1 is based on a constant number of stations.

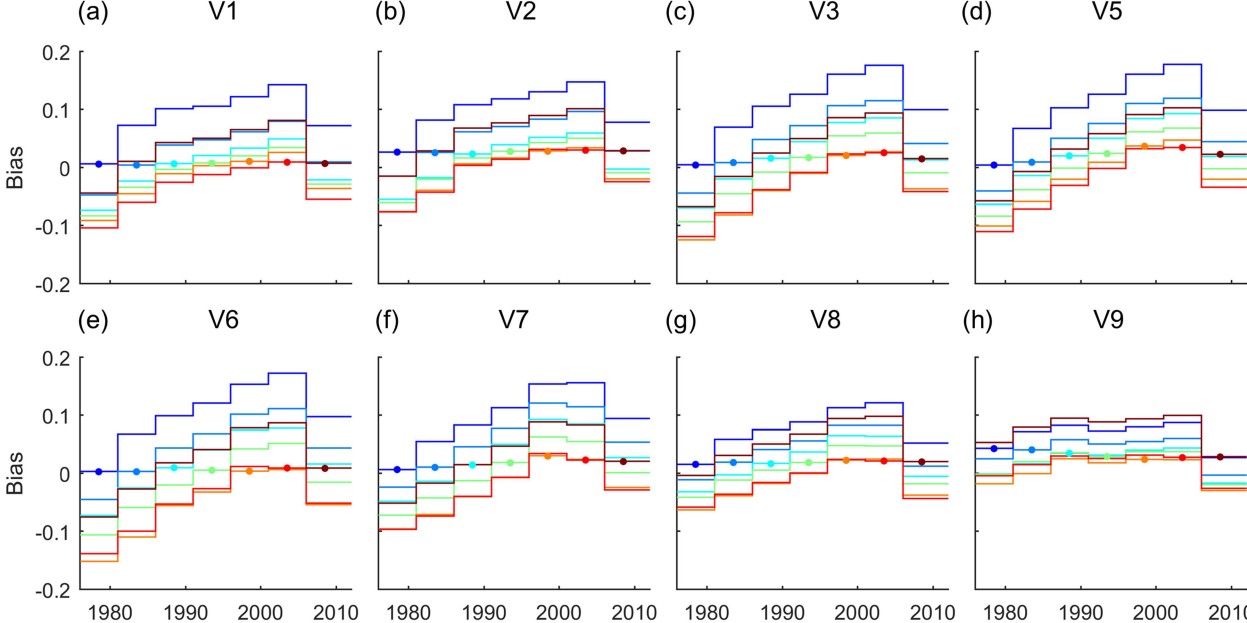

**Figure 5** Bias for the different subperiods averaged over all study catchments for model variants V1–V3 and V5–V9 (model variant V4 was not calibrated for different subperiods). Figure 3a shows this for the baseline model V0. Each line refers to models calibrated in one subperiod, where the filled circle marks the calibration period, showing bias during the calibration period and during evaluation in the other six subperiods. For a description of the model variants see Table 3 and section 2.4.2.

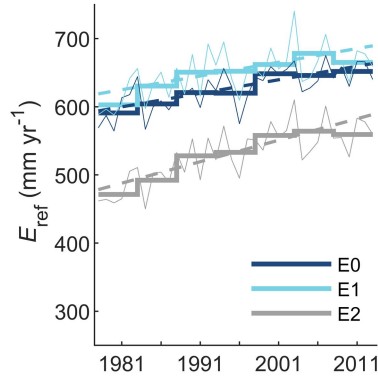

**Figure 6** Temporal variations of $E_{ref}$ as calculated by three different methods, averaged over all catchments. The thick lines show subperiod means, the thin lines annual sums, and the dashed lines linear trends, the different colours represent different data sets. Calculation of $E_{ref}$ by: E0 Blaney-Criddle, E1 Penman-Monteith, E2 Penman-Monteith using a variable surface resistance based on changes in a satellite-based vegetation index.