# Peer review of "Why does a conceptual hydrological model fail to correctly predict discharge changes in response to climate change?"

_Hydrology and Earth System Sciences, 2019_

## Short Comment (SC1) · 7 Jan 2020

This manuscript tries to untangle one of the most challenging problems in hydrology, and it has implications to more than hydrology models: why even a calibrated hydrology model is not reliable for future simulations?

While the authors lay down quite great effort to test and examine some hypothesis, its vision and credibility may be shorten by some major limitations. It is great to see authors went through input driving data (precipitation, temperature, etc.) to all the way up to discharge. The whole analytical process was very convincing.

[Figure]

Regardless of the model details, I only have a couple of concerns and comments. First, various spatially distributed hydrology models were used across scales. The authors need to justify why HBV is representative here. There are models considering vegetation dynamics for example.

Second, as authors pointed out many sources may contribute to model low performance, I suggest there should be at least more evaluations of various hydrological processes. For example, the spatial maps of snow cover, SWE, canopy interception, runoff, snowmelt, soil moisture, etc. A cost function only focus on discharge will likely miss a lot of information. We all know a combination of different parameters can produce the similar results but only one of them is the correct set. The only way to reduce this uncertainty is to examine every single step.

---

## Short Comment (SC2) · 11 Feb 2020

Comments are from the discussion during a workshop by: Yan Liu, Veit Blauhut, Amelie Herzog, Tunde Olarinoye, Ruth Stephan

The study "Why does a conceptual hydrological model fail to predict discharge changes in response to climate change?" by Duethmann et al presents a very interesting topic, which tries to find important factors that influence the prediction capability of conceptual hydrological models, especially under climate change. In this study, the HBV model was used as one representative of conceptual hydrological models. Three aspects regarding precipitation input, model calibration period, and potential evapotransporation

(LAI and NDVI were used to consider changes of vegetation dynamics and land cover) were investigated to discuss the causes why HBV model fails to predict discharge under changing climate. This study is in the scope of HESS and well written. After reading and discussing this manuscript during a workshop, we thought that posting our comments might be helpful for improving the manuscript. We have following major and specific points:

Major points: 1) Title and abstract are a bit misleading because the results are not generalising for all hydrological models but using HBV as one representative. It would be better to explicitly state that the results are based on HBV model in the abstract. Using subtitle may also help clarifying this issue. 2) The prior distribution of model parameters was assumed to be the beta-distribution. In such way, by giving shaping parameters $\alpha$ and $\beta$ for the beta distribution, it seems that the optimal parameter ranges (high probability density part of the beta distribution) are known for the prior. That will affect the model calibration. To justify why using a beta distribution not a uniform distribution for the parameter prior distribution is needed in the method section. 3) Since the results were analyzed for the averages over 156 catchments, it would be better to see the probability density distribution of the bias (Qobs-Qsim) of all catchments for the prediction periods to support that the low predictability of the averages of all catchments is not due to several catchments that bring very big bias. Providing this information in the supplement will strongly support the results.

Specific points: 1) A northern arrow is missing in Fig.1, the elevation legend is normally vertical. Fig. 2 is not very informative, maybe merge it with Fig. 1. 2) In Fig. 4, how was the bias calculated. 3) What does the unit "mm yr-1 per 35 yrs" mean? Is that the mean discharge (mm yr-1) over the 35 years? 4) In equation 8, definition of fbeta is missing. fp was not used. 5) In Sect. 2.3.1, many model parameters were introduced, such as CR and Bmax, but these two parameters are not provided in Table 1. 6) Table 2 contains almost all the details of hypotheses. But there is also quite long text in Sect. 2.4.2 that repeats the table. Table 2 is clear, try to reduce the duplicate text in Sect.

2.4.2. 7) Hypothesis should be a result out of the introduction and be mentioned at the last paragraph in the introduction. 8) In the discussion, very good literature review was done. But it should more highlight the findings of this study and relate and compare to literatures. 9) It is not clear that how the trend was calculated when using 25 years as the calibration period. Please clarify that.

———————————————

---

## Referee Comment (RC1) · Mojca Sraj (Referee) · 14 Feb 2020

General comments

The paper investigates the reasons for failure of the conceptual hydrological model predicting changes in discharge as a response to observed increases in precipitation and air temperature for 156 catchments in Austria. The authors considered three groups of possible causes, namely data problems (precipitation, temperature), problems related to the model calibration (length of calibration period, objective function), and problems of the model structure (ET calculation method, vegetation changes). Hypotheses of the possible causes were evaluated using simulations with modifications of the base-

line model. The paper is in the scope of the journal. It is well written and structured. The data seem to be of appropriate quality. References are up to date and appropriate. There are, however, some areas that require minor corrections for further improvement.

1. Penman-Monteith method – Please, check the calculation and the equation for the net radiation at the crop surface (Eq. 4). Net radiation (Rn) is the difference between the incoming net shortwave radiation (Rns) and the outgoing net longwave radiation (Rnl), and Rns is derived from the balance between incoming and reflected global radiation (Rs) given by $(1-\alpha)$*Rs (see Allen et al., 1998).

Specific comments

1. Page 1, line 21: I would suggest being more specific in defining the size of the impact. How much is "little"? Is it negligible?

2. Page 3, line 29: Using abbreviation Merz2011 for the reference is unconventional. I would suggest to use the usual way of citing, namely Merz et al. (2011). This issue should be corrected throughout the document.

3. Page 7, line 16: Please, check equation 4. It does not seem ok to me.

4. Page 7, line 19: Modified Eref calculated using a variable surface resistance based on changes in a satellite-based vegetation index should be marked as E2 (not E3) in order to be consistent with Table 3.

5. Page 8, lines 9-10: Abbreviations E1 and E2 should be explained when first mentioned. Furthermore, correct the numbering of E1, E2 and E3 to be consistent with Table 3.

6. Page 14, Table 3: It would be useful for readers to include the exact years of the 5-year calibration period in the table or in the table caption. Is it the first 5 years of the considered data or any other? Is it the same for all model variants?

7. Page 15, lines 3-6: Analyses of simulated changes in storage should be explained

in more detail since they are mentioned only in this paragraph.

8. Page 15, lines 13-16: As seen from Fig. 4b, an increase in model performance loss with increasing distance of evaluation periods from the calibration period could be observed in almost all cases, regardless of the calibration period. Please, rewrite the sentence.

9. Page 20, Table 5: It would be useful for readers to add a corresponding model variant to each individual result.

10. Page 21, line 3: Please add which 5 years and 25 years were used for calibration of the mentioned model variants.

11. Page 21, line 21: It would be useful for readers to add a model variant in brackets.

Technical corrections

1. Page 15, line 18: It should be "is reversed".

2. Page 15, line 14: Correct the structure of the sentence.

3. Page 24, line 4: Bracket is missing at the end of the sentence.

---

## Referee Comment (RC2) · Anonymous Referee #2 · 17 Feb 2020

In their manuscript "Why does a conceptual hydrological model fail to predict discharge changes in response to climate change?", D. Duethmann et al. investigate possible reasons for the deficiencies of a conceptual hydrological model (HBV model type) in reproducing observed changes in discharge as a response to changing hydrometeorological conditions in 156 catchments in Austria. The authors set up hypotheses that belong to three groups of possible causes: (i) data problems, (ii) problems related to model calibration, and (iii) problems related to model structure. They test these hypotheses by comparing simulations generated by modified versions of the model according to the hypotheses against a baseline model. Data problems and model structural problems with respect to vegetation dynamics have been identified as the most

relevant causes for the model deficiencies.

General comments:

The paper is well written and well structured. It addresses a relevant scientific question and provides valuable insights for hydrological modelling under changing climate conditions which surely is of broad interest. Still, I have a few comments and suggestions that may further improve the manuscript:

The results are mostly presented as averages over the investigated 156 catchments. I wonder if we could not learn even more if also the statistical and/or spatial distributions will be presented. As stated in the discussion, reasons for hydrological model deficiencies can be very site specific. By including more of the variability between the catchments, prominent cases could be identified which do not (or particularly do) support the conclusions which are based on the mean of all 156 catchments. This may also feed the discussion on possible further causes for model deficiencies which have not been tested in this study.

The modified model versions V2, V7, and V8 have led to the best improvements. Maybe it is worth showing another figure on these results in the same manner as Fig. 3 (or the modified version of Fig. 3). This could be a nice illustration of the key results of this study.

Specific comments:

Title: The title is catchy but also provocative since it suggests that conceptual hydrological models in general are not suited/justified for climate change impact studies, which is not correct.

P2, ll9-11: what is meant by "minimum requirement". Passing or failing the test? How is this determined?

P3, l25: Please provide references.

P4, l14: The numbers show comparatively large differences in elevation ranges. I wonder if this has any influence on the testing result. Are there any altitude-dependent differences in the results of testing the hypothesis? This partly corresponds to my general comment.

P4, Fig.1: When I look at this map, I am reminded to a paper that has identified (homogenous) hydrological regions in Austria (though it was probably with reference to flood types). Anyway, do the presented testing results show any systematic spatial differences regarding the major reasons for model performance losses or improvements? For the baseline model, Fig. 3 (c) presents a map in this regard. For the tested hypotheses, however, spatial information is not presented. I think, though, that this could be interesting. This also corresponds to my general comment.

P5, ll9-13: I remember from other regions and countries that their official meteorological data products are already corrected for potential undercatch. I am not familiar with the SPARTACUS data; I just want to be sure that no "double-correction" is performed here.

P6, Section 2.3.1 could also make a reference to Table 1.

P7, l19, and P8, ll9-10: "(E3)" confuses me. Did I miss E2? On P8, E3 is compared to E2. Later, only results for E0-E2 are reported (e.g. Table 3). I assume that E3 is E2. Please check. Also, "than" instead of "tha" (P8, l9).

P8, Eq.8: Is fbeta the same as fp? Otherwise, fbeta is not explained. Is the same objective function applied in Merz2011?

P9, l4: One more sentence on how the shuffled complex evolution algorithm works would be nice.

P9, l9: It could be highlight that the seven 5-year calibration periods have no temporal overlap.

P10, ll13-16ff: I agree that such problems will probably not affect many catchments.

For selected catchments, particularly in mountainous areas, it still might be a cause for problems in calibrating and evaluation the hydrological model. Does the HZB provide information in this regard?

P11, Figure 2: You may add to the figure caption to which number of stations P1 (P2) and T1 refer.

P14, l18: Does Esim refer to the model estimation based in Eq.2? Or does it refer to the difference between P-Qsim? Would it make any difference (also regarding the consideration of the same uncertainties that refer to the estimation of Ewb)?

P15, ll3-6: How has this been done?

P16, Figure 3 (and others): I see that these figures are designed to match the presentation by Merz2011. However, I think that by presenting only the mean a lot of information is hidden. Boxplots or additional maps (as in Fig. 3) would be more appropriate. This also refers to my general comment.

P17, Figure 4: Do the seven 5-year calibration- and evaluation periods show any marked differences in terms of hydro-meteorological conditions?

P18, Figure 5 (also Figure 7): You could add to the figure caption that the impacts of altering these variants in the hydrological model are summarized in Table 4.

P19, Figure 6: You may indicate that Fig. 6 (a) is the same as Fig. 4 (a).

P20, Table 5: This table (in combination with Table 2) is really nice since it provides a good summary of the tested hypotheses. Maybe the result of V8 can also be summarized here.

P21, ll25-27ff: It could be emphasized more clearly why you choose to combine V2 with V7 to V8.

P22, Discussion: The discussion reads nicely, and I agree with the main conclusion that the consideration of interrelations between climate, vegetation, and hydrology is an

important further step for hydrological modelling in transient climate. Still, I have a few remarks and thoughts regarding the discussion. a) The discussion in its current form gives the impression that model structure deficiencies regarding vegetation dynamics is the most important reason for model performance deficiencies in transient climate, although fixing problems in the precipitation data have led to improvements of similar magnitude. Finally, it could be highlighted that the combination of both approaches has led to the largest improvement (reduction in mismatch by about 95%). b) For good reasons, model structure improvements are restricted to incorporating vegetation dynamics only. Still, what could be further model structural issues that cause model performance losses in this particular study region? Maybe it is worth highlighting that glaciated catchments have not been considered here. Have they been considered by Merz2011?

P23, ll1-3: Considering my complaint regarding the title: This is a good example for the benefit of a conceptual hydrological model. By applying a rather simple approach, vegetation dynamics can be considered to some degree for hydrological simulations in changing climates.

P24, l1: I think this refers to V2 which indeed had a considerable effect.

P24, l4: One ")" is missing.

---

## Short Comment (SC3) · 24 Feb 2020

This study (Duethmann, D., Blöschl, G., and Parajka, J.: Why does a conceptual hydrological model fail to predict discharge changes in response to climate change?, Hydrol. Earth Syst. Sci. Discuss., https://doi.org/10.5194/hess-2019-652, in review, 2020) aims to explain the reasons why many conceptual hydrological models fail to predict non-stationary hydrologic behavior under changing climate. They examined three potential sources of errors within a HBV modeling framework: (1) observational error (or uncertainty), (2) parameter error (due to calibration), and (3) model structural error (mostly deficiency of vegetation dynamics). Using factorial design, they tried to deconvolve each contribution in the long-term deviations between observed and simulated streamflow patterns at the 156 Austrian catchments. I totally agree to a key motivation of this study in that traditional hydrologic modeling has often ignored the importance of vegetation responses to changing climate, which can possibly provide key hydrologic non-stationary components. I write down this comment to reply to a first point raised by Dr. Liu et al. I disagree that these results cannot be applicable to other hydrological models, including both conceptual and distributed ones. Vegetation phenology, longer growing season, and subsequent vegetation growth are regarded as key and universal ecosystem responses to warming, which have great implications in carbon and water cycles. However, to my knowledge, few studies have considered these potential feedbacks between vegetation, climate, and hydrology especially in future hydrological modeling. Interactions between vegetation and hydrology can be particularly important in the watershed systems where vegetation dynamics and its water use are strongly coupled with subsequent hydrologic behavior (e.g. forested watersheds). I think that this study would provide timely information and a common ground for hydrologists why vegetation phenology and subsequent dynamics and responses should be included in future hydrological modeling under changing climate.

Taehee Hwang Assistant Professor Department of Geography Indiana University Bloomington

---

## Referee Comment (RC3) · David Post (Referee) · 26 Feb 2020

This is a nice example of a study that attempts to determine exactly what it is about rainfall-runoff models that means they are not capable of predicting well runoff under changed climate conditions. One major thing that would improve the paper would be to quantify for the reader what the change in relevant hydroclimatological characteristics during the verification period actually are. The authors state that the area was subject to significant climate changes, but do not tell us what these actually were. Were the evaluation periods drier/hotter? If so, by how much. What were the relative runoff coefficients?

[Figure]

Despite these issues, I have just three comments on improving the paper:

1. The title is misleading. Almost every model will predict discharge changes in response to climate change. The questions is why they do not 'accurately' predict discharge changes? The addition of a qualifier like 'accurately' would be useful.

2. Changes in anthropogenic influences are largely ignored as the authors claim that the catchments are largely unregulated and existing diversions were introduced before the beginning of the study period. I would question this. While the diversions may be in place before the beginning of the study period, are there operating rules related to this diversions which may vary from year to year, for example allowing larger diversions during periods of low flow (or vice-versa). I ask as we have identified catchments in Australia that not only behaved abnormally (gave lower than predicted yields during the Millennium drought), but that have not returned to 'normal' yields post-drought. One hypothesis for this is that farmers sank groundwater bores to access an alternative water supply during the drought when they were unable to pump from surface water. Any lowering of the groundwater table resulting from this activity would obviously lead to lower than expected yields. Once this 'sunk cost' had been incurred, there would be no benefit to farmers in ceasing the pumping of water from these bores, thus they may still be doing so post-drought. Such anthropogenic influences are of course hard to determine (and even harder to quantify), but the authors would do well to keep them in mind.

3. The assessment that problems with the model calibration can be the source of the poor performance during the evaluation period is a good one. In particular, that processes that are relevant in the calibration period are not present (or 'activated' to use the author's terminology) in the calibration period. I am not sure that extending the calibration period from 5 to 25 years will actually evaluate whether this is the case. It may be that these processes will be seen in the 25 year period, but it may not. One thing that could be done is to compare the model that is calibrated on the evaluation period (or perhaps part of it) to the model that is calibrated on the calibration period.

If different processes are dominant in the evaluation period, this would be seen in how these models perform on an independent data set.

---

## Author Comment (AC2) · 30 Mar 2020

**Replies to the comments by Referee #2**

We would like to thank the anonymous referee for his/her interest and the comments on our manuscript.

Below, reviewer comments are in italic font and our replies are in normal font.

*In their manuscript "Why does a conceptual hydrological model fail to predict discharge changes in response to climate change?", D. Duethmann et al. investigate possible reasons for the deficiencies of a conceptual hydrological model (HBV model type) in reproducing observed changes in discharge as a response to changing hydrometeorological conditions in 156 catchments in Austria. The authors set up hypotheses that belong to three groups of possible causes: (i) data problems, (ii) problems related to model calibration, and (iii) problems related to model structure. They test these hypotheses by comparing simulations generated by modified versions of the model according to the hypotheses against a baseline model. Data problems and model structural problems with respect to vegetation dynamics have been identified as the most relevant causes for the model deficiencies.*

*General comments:*

*The paper is well written and well structured. It addresses a relevant scientific question and provides valuable insights for hydrological modelling under changing climate conditions which surely is of broad interest. Still, I have a few comments and suggestions that may further improve the manuscript:*

*The results are mostly presented as averages over the investigated 156 catchments. I wonder if we could not learn even more if also the statistical and/or spatial distributions will be presented. As stated in the discussion, reasons for hydrological model deficiencies can be very site specific. By including more of the variability between the catchments, prominent cases could be identified which do not (or particularly do) support the conclusions which are based on the mean of all 156 catchments. This may also feed the discussion on possible further causes for model deficiencies which have not been tested in this study. The modified model versions V2, V7, and V8 have led to the best improvements. Maybe it is worth showing another figure on these results in the same manner as Fig. 3 (or the modified version of Fig. 3). This could be a nice illustration of the key results of this study.*

> We agree with the reviewer that many details are hidden by aggregating the results to annual means over all catchments. While we need to aggregate the results to a large extent due to the large amount of data, we will show more spatial patterns and distributions across catchments in the revised manuscript. In particular, we will include maps similar to Fig. 3 (c) for selected model variants as suggested by the reviewer. We will further show distributions across the 156 study catchments of bias and NSE for the baseline model calibrated in the different subperiods, complementing Fig. 4 that shows changes in the mean value of bias and NSE averaged across the study catchments, as suggested by Yan Liu, Veit Blauhut, Amelie Herzog, Tunde Olarinoye and Ruth Stephan (second short comment).

*Specific comments:*

*Title: The title is catchy but also provocative since it suggests that conceptual hydrological models in general are not suited/justified for climate change impact studies, which is not correct.*

> We will revise the title, please refer to the comment #1 by David Post. The new title reads 'Why does a conceptual hydrological model fail to correctly predict discharge changes in response to climate change?' By referring to 'a conceptual hydrological model' and not 'conceptual hydrological models' we intend to indicate that we have tested one and not several or more models. The problems we found when applying the HBV-based model over catchments in Austria may, however, also be relevant for other hydrological models and other regions (also see SC3 by Taehee Hwang).

*P2, ll9-11: what is meant by "minimum requirement". Passing or failing the test? How is this determined?*

> Passing the DSST can be seen as a minimum requirement for models applied for climate impact assessments. The text will be adjusted to make this clearer.

*P3, l25: Please provide references.*

> Will be added to the manuscript (Fowler et al., 2018;Fowler et al., 2016;Westra et al., 2014).

*P4, l14: The numbers show comparatively large differences in elevation ranges. I wonder if this has any influence on the testing result. Are there any altitude-dependent differences in the results of testing the hypothesis? This partly corresponds to my general comment.*

> The relationship between catchment elevation and the trend of the gap between simulated minus observed discharge is not very conclusive (Fig. 1). On the one hand, catchments in the lowest elevation class (median catchment elevations below 400 m) show clearly lower deviations between simulated and observed trends. Furthermore, there is a slight increase of the gap between simulated and observed trends with elevation for median catchment elevations up to 1200 m. On the other hand, this tendency largely disappears when the gap between simulated and observed trends is normalized by the mean annual observed discharge, and the group of catchments with median elevations below 400 m is based on only 7 of the 156 catchments.

[Figure]

**Figure 1** (a) Boxplots of the differences of simulated minus observed trends in discharge against median catchment elevation and (b) boxplots of the differences of simulated minus observed trends in discharge against median catchment elevation normalized by average annual discharge.

*P4, Fig.1: When I look at this map, I am reminded to a paper that has identified (homogenous) hydrological regions in Austria (though it was probably with reference to flood types). Anyway, do the presented testing results show any systematic spatial differences regarding the major reasons for model performance losses or improvements? For the baseline model, Fig. 3 (c) presents a map in this regard. For the tested hypotheses, however, spatial information is not presented. I think, though, that this could be interesting. This also corresponds to my general comment.*

We will show maps similar to Fig. 3 (c) for model variants V2, V7 and V8, as suggested by the reviewer in the general comments. This shows that using the precipitation data set P2 resulted in reduced gaps between trends of simulated minus observed discharge particularly for catchments with large trends in simulated minus observed discharge, whereas considering vegetation dynamics for the calculation of evapotranspiration resulted in a much more even effect between catchments. V8 combines both of these effects, reducing the trend of simulated minus observed discharge in most catchments with large reductions in catchments that showed large trends of simulated minus observed discharge in the baseline model.

*P5, ll9-13: I remember from other regions and countries that their official meteorological data products are already corrected for potential undercatch. I am not familiar with the SPARTACUS data; I just want to be sure that no "double-correction" is performed here.*

The SPARTACUS data are not corrected for undercatch (Hiebl and Frei, 2017).

*P6, Section 2.3.1 could also make a reference to Table 1.*

Will be added.

*P7, l19, and P8, ll9-10: "(E3)" confuses me. Did I miss E2? On P8, E3 is compared to E2. Later, only results for E0-E2 are reported (e.g. Table 3). I assume that E3 is E2. Please check. Also, "than" instead of "tha" (P8, l9).*

Thanks for pointing this out. This will be corrected.

*P8, Eq.8: Is fbeta the same as fp? Otherwise, fbeta is not explained. Is the same objective function applied in Merz2011?*

Thank you, yes this is the same and this will be corrected in the manuscript. In this study, we added a penalty for the volume bias in order to keep it low, which was not considered by Merz2011.

*P9, l4: One more sentence on how the shuffled complex evolution algorithm works would be nice.*

Ok, we will add a short explanation.

*P9, l9: It could be highlight that the seven 5-year calibration periods have no temporal overlap.*

Will be added.

*P10, ll13-16ff: I agree that such problems will probably not affect many catchments. For selected catchments, particularly in mountainous areas, it still might be a cause for problems in calibrating and evaluation the hydrological model. Does the HZB provide information in this regard?*

Information on abstractions and flow diversions is provided in the hydrological yearbooks (BMLFUW, 2015). Catchments where flow diversions were introduced before the beginning of the study period were included in the data set, since we did not expect large effects on simulated discharge trends. We excluded catchments where diversions were introduced during the study period.

*P11, Figure 2: You may add to the figure caption to which number of stations P1 (P2) and T1 refer.*

The data sets P1, P2 and T1 are based on a constant number of station series that extend over the entire period. The number of stations they refer to will be added to the text.

*P14, l18: Does Esim refer to the model estimation based in Eq.2? Or does it refer to the difference between P-Qsim? Would it make any difference (also regarding the consideration of the same uncertainties that refer to the estimation of Ewb)?*

The calculation of $E_{sim}$ is described in Section 2.3.1 (P6, L16ff). For the baseline model, $E_{ref}$ is calculated using Eq. 2. $E_{sim}$ is then calculated as a function of $E_{ref}$ and soil moisture. Thus, $E_{wb}$ also includes storage changes, wheras $E_{sim}$ does not. This will be pointed out in the manuscript. This difference is relevant at short time scales. For example, the large year-to-year variations of $E_{wb}$ in Fig. 3 (b) are likely due to storage changes. The mean values over a 5-years subperiod and the trend over the entire study period is much less influenced by any storage changes. We will add more explanation to the text. We will also add an additional figure that shows the differences between precipitation minus runoff for observations and simulations to the supplement.

*P15, ll3-6: How has this been done?*

We will add some more information on how we calculate changes in simulated storage.

"For this, we analysed the sum of all simulated storages, i.e. soil moisture store, upper zone and lower zone groundwater store and snow water equivalent, and calculated trends of annually average values (based on hydrological years). Trends in simulated storage changes were, on average over all catchments, 8 ± 20 mm over 1978–2013. This shows that the overestimation of the discharge trend is not generated by an opposite trend in a storage component. Of the simulated storage groundwater is the largest component. Small changes in simulated storage are also in agreement with no consistent large scale groundwater changes in the observations (Blaschke et al., 2011; Neunteufel et al., 2017)."

*P16, Figure 3 (and others): I see that these figures are designed to match the presentation by Merz2011. However, I think that by presenting only the mean a lot of information is hidden. Boxplots or additional maps (as in Fig. 3) would be more appropriate. This also refers to my general comment.*

We will add further maps similar to Fig. 3 (c) for selected model variants, as suggested by the reviewer in the general comments. We will further add violin plots showing distributions of the bias and NSE complementary to Fig. 4.

*P17, Figure 4: Do the seven 5-year calibration- and evaluation periods show any marked differences in terms of hydro-meteorological conditions?*

Yes. Over the study period, precipitation, air temperature and $E_{ref}$ increased, as shown in Fig. 5 (a–b) and Fig. 7. We will add a description of the changes in the hydro-meteorological conditions to Section 2.1.

*P18, Figure 5 (also Figure 7): You could add to the figure caption that the impacts of altering these variants in the hydrological model are summarized in Table 4.*

Results of these model variants are further also shown in Table 5 and Fig. 6. We will think of adding cross-references to the figure captions.

*P19, Figure 6: You may indicate that Fig. 6 (a) is the same as Fig. 4 (a).*

Will be added.

*P20, Table 5: This table (in combination with Table 2) is really nice since it provides a good summary of the tested hypotheses. Maybe the result of V8 can also be summarized here.*

We did not include V8 in Table 2 or Table 5 because it was not part of the original set of hypotheses. However, we will provide the same information that we provide for the other model variants in Table 5 in the text (where it is missing in the current version).

*P21, ll25-27ff: It could be emphasized more clearly why you choose to combine V2 with V7 to V8.*

> Ok, will be added.

*P22, Discussion: The discussion reads nicely, and I agree with the main conclusion that the consideration of interrelations between climate, vegetation, and hydrology is an important further step for hydrological modelling in transient climate. Still, I have a few remarks and thoughts regarding the discussion.*

*a) The discussion in its current form gives the impression that model structure deficiencies regarding vegetation dynamics is the most important reason for model performance deficiencies in transient climate, although fixing problems in the precipitation data have led to improvements of similar magnitude. Finally, it could be highlighted that the combination of both approaches has led to the largest improvement (reduction in mismatch by about 95%).*

> Thanks for the feedback; it was not our intention to give this impression. We will adjust the discussion to avoid giving this impression. We will also pick up the results of combining the modifications for the precipitation data and considering vegetation dynamics.

*b) For good reasons, model structure improvements are restricted to incorporating vegetation dynamics only. Still, what could be further model structural issues that cause model performance losses in this particular study region? Maybe it is worth highlighting that glaciated catchments have not been considered here. Have they been considered by Merz2011?*

> Good point. We will mention possible model structural problems with respect to changes in glacier extent and glacier volume in the discussion.

*P23, ll1-3: Considering my complaint regarding the title: This is a good example for the benefit of a conceptual hydrological model. By applying a rather simple approach, vegetation dynamics can be considered to some degree for hydrological simulations in changing climates.*

> Please see our comments regarding the title above. While we tried to include changes in vegetation dynamics into a conceptual hydrological model in this study in order to derive a first order estimate of the possible effects, changes in vegetation dynamics are not considered by most conceptual hydrological models.

*P24, l1: I think this refers to V2 which indeed had a considerable effect.*

> V2 had a considerable effect when compared to V0. However, it builds on V1 and the differences between V2 and V1 are small and not significant. This will be clarified in the manuscript.

*P24, l4: One ")" is missing.*

Will be corrected.

**References**

BMLFUW: Hydrographisches Jahrbuch von Österreich 2013, 121. Band - Daten und Auswertungen, Wien, 2015.

Fowler, K., Coxon, G., Freer, J., Peel, M., Wagener, T., Western, A., Woods, R., and Zhang, L.: Simulating Runoff Under Changing Climatic Conditions: A Framework for Model Improvement, Water Resour. Res., 54, 9812-9832, 10.1029/2018wr023989, 2018.

Fowler, K. J. A., Peel, M. C., Western, A. W., Zhang, L., and Peterson, T. J.: Simulating runoff under changing climatic conditions: Revisiting an apparent deficiency of conceptual rainfall-runoff models, Water Resour. Res., 52, 1820-1846, 10.1002/2015wr018068, 2016.

Hiebl, J., and Frei, C.: Daily precipitation grids for Austria since 1961—development and evaluation of a spatial dataset for hydroclimatic monitoring and modelling, Theor. Appl. Climatol., 10.1007/s00704-017-2093-x, 2017.

Westra, S., Thyer, M., Leonard, M., Kavetski, D., and Lambert, M.: A strategy for diagnosing and interpreting hydrological model nonstationarity, Water Resour. Res., 50, 5090-5113, 10.1002/2013wr014719, 2014.

---

## Author Comment (AC6) · 30 Mar 2020

We would like to thank Taehee Hwang for posting his entirely positive comment on our paper. Taehee Hwang replied to a comment by Dr. Liu. He underlined the relevance of our study and emphasized the problem that many hydrologic models neglect changes in vegetation dynamics even though vegetation responses to climate change may have important influences on hydrologic systems.

---

## Author Response (AR1)

**Authors' response - Manuscript "Why does a conceptual hydrological model fail to correctly predict discharge changes in response to climate change?" by D. Duethmann, G. Blöschl and J. Parajka**

**Replies to the comments by Mojca Sraj**

We would like to thank Mojca Sraj for her interest and for her comments on our manuscript.

Below, reviewer comments are in italic font and our replies are in normal font.

*General comments*

*The paper investigates the reasons for failure of the conceptual hydrological model predicting changes in discharge as a response to observed increases in precipitation and air temperature for 156 catchments in Austria. The authors considered three groups of possible causes, namely data problems (precipitation, temperature), problems related to the model calibration (length of calibration period, objective function), and problems of the model structure (ET calculation method, vegetation changes). Hypotheses of the possible causes were evaluated using simulations with modifications of the baseline model. The paper is in the scope of the journal. It is well written and structured. The data seem to be of appropriate quality. References are up to date and appropriate. There are, however, some areas that require minor corrections for further improvement.*

*1. Penman-Monteith method – Please, check the calculation and the equation for the net radiation at the crop surface (Eq. 4). Net radiation (Rn) is the difference between the incoming net shortwave radiation (Rns) and the outgoing net longwave radiation (Rnl), and Rns is derived from the balance between incoming and reflected global radiation (Rs) given by (1-\_)\*Rs (see Allen et al., 1998).*

> Thank you very much for pointing this out. This will be corrected. The error only occurred in the manuscript (not in the calculations).

*Specific comments*

*1. Page 1, line 21: I would suggest being more specific in defining the size of the impact. How much is "little"? Is it negligible?*

> We will add "(less than 5 mm $yr^{-1}$ per 35 yrs)" to be more specific.

*2. Page 3, line 29: Using abbreviation Merz2011 for the reference is unconventional. I would suggest to use the usual way of citing, namely Merz et al. (2011). This issue should be corrected throughout the document.*

> Since Merz et al. (2011) is referred to very often, it seems a useful abbreviation and we will adjust this according to any guidance by the journal.

*3. Page 7, line 16: Please, check equation 4. It does not seem ok to me.*

See above.

*4. Page 7, line 19: Modified Eref calculated using a variable surface resistance based on changes in a satellite-based vegetation index should be marked as E2 (not E3) in order to be consistent with Table 3.*

Thank you for pointing this out. This will be changed.

*5. Page 8, lines 9-10: Abbreviations E1 and E2 should be explained when first mentioned. Furthermore, correct the numbering of E1, E2 and E3 to be consistent with Table 3.*

E0 and E1 are defined in Eq. 2 and Eq. 3. The numbering will be corrected to be consistent with Table 3.

*6. Page 14, Table 3: It would be useful for readers to include the exact years of the 5-year calibration period in the table or in the table caption. Is it the first 5 years of the considered data or any other? Is it the same for all model variants?*

Model calibration and the calibration periods are described in Section 2.3.3. We used seven 5 year calibration periods (based on hydrological years), during 1978–2012. As a modification, we also tested using a 25-year period as calibration period (1978–2002). We will change the header of the respective column in Table 3 from "Calibration period" to "Length of calibration periods".

*7. Page 15, lines 3-6: Analyses of simulated changes in storage should be explained in more detail since they are mentioned only in this paragraph.*

We will add some more information on how we calculate changes in simulated storage and will extend this paragraph in the manuscript.

"For this, we analysed the sum of all simulated storages, i.e. soil moisture store, upper and lower zone subsurface store and snow water equivalent, and calculated trends of annually average values (based on hydrological years). Trends in simulated storage changes were, on average over all catchments, 8 ± 20 mm over 1978–2013. This shows that the overestimation of the discharge trend is not generated by an opposite trend in simulated storage. Small changes in simulated storage are in agreement with no consistent large scale groundwater changes in the observations (Blaschke et al., 2011; Neunteufel et al., 2017). "

*8. Page 15, lines 13-16: As seen from Fig. 4b, an increase in model performance loss with increasing distance of evaluation periods from the calibration period could be observed in almost all cases, regardless of the calibration period. Please, rewrite the sentence.*

Will be rewritten: "In many cases, model performance decreases with increasing distance between the calibration and the evaluation period, particularly for model evaluations in subperiod S1 and S2."

*9. Page 20, Table 5: It would be useful for readers to add a corresponding model variant to each individual result.*

Good idea, will be added.

*10. Page 21, line 3: Please add which 5 years and 25 years were used for calibration of the mentioned model variants.*

Will be added.

*11. Page 21, line 21: It would be useful for readers to add a model variant in brackets.*

Will be added.

*Technical corrections*

*1. Page 15, line 18: It should be "is reversed".*

(Note, the comment apparently refers to page 14, line 18). Ok, can be changed.

*2. Page 15, line 14: Correct the structure of the sentence.*

See above, we will rewrite the sentence. "In many cases, model performance decreases with increasing distance between the calibration and the evaluation period, particularly for model evaluation in subperiod S1 and S2."

*3. Page 24, line 4: Bracket is missing at the end of the sentence.*

Thanks, will be corrected.
* * *
**Replies to the comments by Referee #2**

We would like to thank the anonymous referee for his/her interest and the comments on our manuscript.

Below, reviewer comments are in italic font and our replies are in normal font.

*In their manuscript "Why does a conceptual hydrological model fail to predict discharge changes in response to climate change?", D. Duethmann et al. investigate possible reasons for the deficiencies of a conceptual hydrological model (HBV model type) in reproducing observed changes in discharge as a response to changing hydrometeorological conditions in 156 catchments in Austria. The authors set up hypotheses that belong to three groups of possible causes: (i) data problems, (ii) problems related to model calibration, and (iii) problems related to model structure. They test these hypotheses by comparing simulations generated by modified versions of the model according to the hypotheses*

*against a baseline model. Data problems and model structural problems with respect to vegetation dynamics have been identified as the most relevant causes for the model deficiencies.*

*General comments:*

*The paper is well written and well structured. It addresses a relevant scientific question and provides valuable insights for hydrological modelling under changing climate conditions which surely is of broad interest. Still, I have a few comments and suggestions that may further improve the manuscript:*

*The results are mostly presented as averages over the investigated 156 catchments. I wonder if we could not learn even more if also the statistical and/or spatial distributions will be presented. As stated in the discussion, reasons for hydrological model deficiencies can be very site specific. By including more of the variability between the catchments, prominent cases could be identified which do not (or particularly do) support the conclusions which are based on the mean of all 156 catchments. This may also feed the discussion on possible further causes for model deficiencies which have not been tested in this study. The modified model versions V2, V7, and V8 have led to the best improvements. Maybe it is worth showing another figure on these results in the same manner as Fig. 3 (or the modified version of Fig. 3). This could be a nice illustration of the key results of this study.*

> We agree with the reviewer that many details are hidden by aggregating the results to annual means over all catchments. While we need to aggregate the results to a large extent due to the large amount of data, we will show more spatial patterns and distributions across catchments in the revised manuscript. In particular, we will include maps similar to Figure 2c for selected model variants as suggested by the reviewer as Supplementary Figure S6a-c. We will further show distributions across the 156 study catchments of bias and NSE for the baseline model calibrated in the different subperiods as Supplementary Figures 4 and 5, complementing Figure 3 that shows changes in the mean value of bias and NSE averaged across the study catchments, as suggested by Yan Liu, Veit Blauhut, Amelie Herzog, Tunde Olarinoye and Ruth Stephan (second short comment).

*Specific comments:*

*Title: The title is catchy but also provocative since it suggests that conceptual hydrological models in general are not suited/justified for climate change impact studies, which is not correct.*

> We will revise the title, please refer to the comment #1 by David Post. The new title reads 'Why does a conceptual hydrological model fail to correctly predict discharge changes in response to climate change?' By referring to 'a conceptual hydrological model' and not 'conceptual hydrological models' we intend to indicate that we have tested one and not several or more models. The problems we found when applying the HBV-based model over catchments in Austria may, however, also be relevant for other hydrological models and other regions (also see SC3 by Taehee Hwang).

*P2, ll9-11: what is meant by "minimum requirement". Passing or failing the test? How is this determined?*

Passing the DSST can be seen as a minimum requirement for models applied for climate impact assessments. The text will be adjusted to make this clearer.

*P3, l25: Please provide references.*

Will be added to the manuscript (Fowler et al., 2018;Fowler et al., 2016;Westra et al., 2014).

*P4, l14: The numbers show comparatively large differences in elevation ranges. I wonder if this has any influence on the testing result. Are there any altitude-dependent differences in the results of testing the hypothesis? This partly corresponds to my general comment.*

The relationship between catchment elevation and the trend of the gap between simulated and observed discharge is not very conclusive (Figure 1). On the one hand, catchments in the lowest elevation class (median catchment elevations below 400 m) show clearly lower deviations between simulated and observed trends. Furthermore, there is a slight increase of the gap between simulated and observed trends with elevation for median catchment elevations up to 1200 m. On the other hand, this tendency largely disappears when the gap between simulated and observed trends is normalized by the mean annual observed discharge, and the group of catchments with median elevations below 400 m is based on only 7 of the 156 catchments.

(a)                                              (b)

[Figure]

**Figure 1** (a) Boxplots of the differences of simulated and observed trends in discharge against median catchment elevation and (b) boxplots of the differences of simulated and observed trends in discharge against median catchment elevation normalized by average annual discharge.

*P4, Fig.1: When I look at this map, I am reminded to a paper that has identified (homogenous) hydrological regions in Austria (though it was probably with reference to flood types). Anyway, do the presented testing results show any systematic spatial differences regarding the major reasons for model performance losses or improvements? For the baseline model, Fig. 3 (c) presents a map in this regard. For the tested hypotheses, however, spatial information is not presented. I think, though, that this could be interesting. This also corresponds to my general comment.*

We will show maps similar to Figure 2c for model variants V2, V8 and V9 (V2, V7 and V8 in the original manuscript) as Supplementary Figure S6 (as suggested by the reviewer in the general

comments). This shows that using the precipitation data set P2 resulted in reduced gaps between trends of simulated and observed discharge particularly for catchments with large trends in simulated minus observed discharge, whereas considering vegetation dynamics for the calculation of evapotranspiration resulted in a much more even effect between catchments. V9 combines both of these effects, reducing the trend of simulated minus observed discharge in most catchments with large reductions in catchments that showed large trends of simulated minus observed discharge in the baseline model.

*P5, ll9-13: I remember from other regions and countries that their official meteorological data products are already corrected for potential undercatch. I am not familiar with the SPARTACUS data; I just want to be sure that no "double-correction" is performed here.*

The SPARTACUS data are not corrected for undercatch (Hiebl and Frei, 2017).

*P6, Section 2.3.1 could also make a reference to Table 1.*

Will be added.

*P7, l19, and P8, ll9-10: "(E3)" confuses me. Did I miss E2? On P8, E3 is compared to E2. Later, only results for E0-E2 are reported (e.g. Table 3). I assume that E3 is E2. Please check. Also, "than" instead of "tha" (P8, l9).*

Thanks for pointing this out. This will be corrected.

*P8, Eq.8: Is fbeta the same as fp? Otherwise, fbeta is not explained. Is the same objective function applied in Merz2011?*

Thank you, yes this is the same and this will be corrected in the manuscript. In this study, we added a penalty for the volume bias in order to keep it low, which was not considered by Merz2011.

*P9, l4: One more sentence on how the shuffled complex evolution algorithm works would be nice.*

Ok, we will add a short explanation.

*P9, l9: It could be highlight that the seven 5-year calibration periods have no temporal overlap.*

Will be added.

*P10, ll13-16ff: I agree that such problems will probably not affect many catchments. For selected catchments, particularly in mountainous areas, it still might be a cause for problems in calibrating and evaluation the hydrological model. Does the HZB provide information in this regard?*

Information on abstractions and flow diversions is provided in the hydrological yearbooks (BMLFUW, 2015). Catchments where flow diversions were introduced before the beginning of the study period were included in the data set, since we did not expect large effects on

simulated discharge trends. We excluded catchments where diversions were introduced during the study period.

*P11, Figure 2: You may add to the figure caption to which number of stations P1 (P2) and T1 refer.*

The data sets P1, P2 and T1 are based on a constant number of station series that extend over the entire period. The number of stations they refer to will be added to the text.

*P14, l18: Does Esim refer to the model estimation based in Eq.2? Or does it refer to the difference between P-Qsim? Would it make any difference (also regarding the consideration of the same uncertainties that refer to the estimation of Ewb)?*

The calculation of $E_{sim}$ is described in Section 2.3.1. For the baseline model, $E_{ref}$ is calculated using Eq. 2. $E_{sim}$ is then calculated as a function of $E_{ref}$ and soil moisture. Thus, $E_{wb}$ also includes storage changes, wheras $E_{sim}$ does not. This will be pointed out in the manuscript. This difference is relevant at short time scales. For example, the large year-to-year variations of $E_{wb}$ in Figure 2b are likely due to storage changes. The mean values over a 5-year subperiod and the trend over the entire study period is much less influenced by any storage changes. We will add more explanation to the text. We will also add an additional figure that shows the differences between precipitation and runoff for observations and simulations to the supplement (Supplementary Figure S3).

*P15, ll3-6: How has this been done?*

We will add some more information on how we calculate changes in simulated storage.

"For this, we analysed the sum of all simulated storages, i.e. soil moisture store, upper zone and lower zone groundwater store and snow water equivalent, and calculated trends of annually average values (based on hydrological years). Trends in simulated storage changes were, on average over all catchments, 8 ± 20 mm over 1978–2013. This shows that the overestimation of the discharge trend is not generated by an opposite trend in a storage component. Small changes in simulated storage are also in agreement with no consistent large scale groundwater changes in the observations (Blaschke et al., 2011; Neunteufel et al., 2017)."

*P16, Figure 3 (and others): I see that these figures are designed to match the presentation by Merz2011. However, I think that by presenting only the mean a lot of information is hidden. Boxplots or additional maps (as in Fig. 3) would be more appropriate. This also refers to my general comment.*

We will add further maps similar to Figure 3c for selected model variants, as suggested by the reviewer in the general comments, to the supplement (Supplementary Figure S6). We will further show violin plots with distributions of the bias and NSE in Supplementary Figures S4–S5 complementary to Figure 3.

*P17, Figure 4: Do the seven 5-year calibration- and evaluation periods show any marked differences in terms of hydro-meteorological conditions?*

Yes. Over the study period, precipitation, air temperature and $E_{ref}$ increased, as shown in Figure 4 a–b and Figure 6. We will add a description of the changes in the hydro-meteorological conditions to Section 2.1.

*P18, Figure 5 (also Figure 7): You could add to the figure caption that the impacts of altering these variants in the hydrological model are summarized in Table 4.*

Yes, this could be an idea. In the end, it was not added as there would be more information needed (i.e. which model variant links to which precipitation or air temperature data set), and because this is just one out of many possible cross references between the figures and tables (and adding all of them seems a bit much).

*P19, Figure 6: You may indicate that Fig. 6 (a) is the same as Fig. 4 (a).*

Figure 5a (6a in the original manuscript) has changed and we added a reference to Figure 3a (Figure 4a in the original manuscript).

*P20, Table 5: This table (in combination with Table 2) is really nice since it provides a good summary of the tested hypotheses. Maybe the result of V8 can also be summarized here.*

We did not include V9 (V8 in the original manuscript) in Table 2 or Table 5 because it was not part of the original set of hypotheses. However, we will provide the same information that we provide for the other model variants in Table 5 in the text (where it was missing in the original version).

*P21, ll25-27ff: It could be emphasized more clearly why you choose to combine V2 with V7 to V8.*

Ok, will be added.

*P22, Discussion: The discussion reads nicely, and I agree with the main conclusion that the consideration of interrelations between climate, vegetation, and hydrology is an important further step for hydrological modelling in transient climate. Still, I have a few remarks and thoughts regarding the discussion.*

*a) The discussion in its current form gives the impression that model structure deficiencies regarding vegetation dynamics is the most important reason for model performance deficiencies in transient climate, although fixing problems in the precipitation data have led to improvements of similar magnitude. Finally, it could be highlighted that the combination of both approaches has led to the largest improvement (reduction in mismatch by about 95%).*

Thanks for the feedback; it was not our intention to give this impression. We will adjust the discussion to avoid giving this impression. We will also pick up the results of combining the modifications for the precipitation data and considering vegetation dynamics (see first paragraph in '4 Discussion').

*b) For good reasons, model structure improvements are restricted to incorporating vegetation dynamics only. Still, what could be further model structural issues that cause model performance losses in this particular study region? Maybe it is worth highlighting that glaciated catchments have not been considered here. Have they been considered by Merz2011?*

Good point. We will mention possible model structural problems with respect to changes in glacier extent and glacier volume in the discussion.

"Changes in glacier volume may cause deviations between simulated and observed discharge trends if not accounted for by the model. Therefore, glacier covered catchments were excluded in our study. Model structural deficits with respect to glacier dynamics may be responsible for further deviations between simulated and observed discharge trends in the study by Merz2011, which did not exclude glacier covered catchments, although the total glacier cover of Austria is small (0.5 %; Fischer et al. (2015))."

*P23, ll1-3: Considering my complaint regarding the title: This is a good example for the benefit of a conceptual hydrological model. By applying a rather simple approach, vegetation dynamics can be considered to some degree for hydrological simulations in changing climates.*

Please see our comments regarding the title above. While we tried to include changes in vegetation dynamics into a conceptual hydrological model in this study in order to derive a first order estimate of the possible effects, changes in vegetation dynamics are not considered by most conceptual hydrological models.

*P24, l1: I think this refers to V2 which indeed had a considerable effect.*

V2 had a considerable effect when compared to V0. However, it builds on V1 and the differences between V2 and V1 are small and not significant. This will be clarified in the manuscript (we added "when compared to the simulation using the same precipitation data without undercatch correction.").

P24, l4: One ")" is missing.

Will be corrected.

* * *
**Replies to the comments by David Post**

We would like to thank David Post for his generally positive comment and for his suggestions for improving our manuscript.

Below, reviewer comments are in italic font and our replies are in normal font.

*This is a nice example of a study that attempts to determine exactly what it is about rainfall-runoff models that means they are not capable of predicting well runoff under changed climate conditions. One major thing that would improve the paper would be to quantify for the reader what the change in relevant hydroclimatological characteristics during the verification period actually are. The authors state that the area was subject to significant climate changes, but do not tell us what these actually were. Were the evaluation periods drier/hotter? If so, by how much. What were the relative runoff coefficients?*

> Thanks for this advice. Over the study period, precipitation, air temperature and $E_{ref}$ increased, as shown in Figure 4a–b and Figure 6 (Figure 5a–b and Figure 7 in the original manuscript). We will add a description of the changes in the hydro-meteorological conditions to Section 2.1.

> "Over the period 1977–2014 annual precipitation increased by 32 ± 23 mm yr$^{-1}$ or 2.4 ± 1.7 % per decade (based on undercatch corrected SPARTACUS data), air temperature increased by 0.45 ± 0.09 °C per decade and global radiation increased by 5.1 ± 0.9 W m$^{-2}$ per decade on average over the study catchments. In contrast, discharge did not show strong trends and the average trend over the study period was 0.2 ± 3.1% per decade (Duethmann and Blöschl, 2018)."

> Relative annual runoff coefficients (based on observed discharge data and undercatch corrected SPARTACUS precipitation data) vary in the range of 0.22 and 0.86 between the study catchments. The runoff coefficients significantly decreased ($p \leq 0.05$) in 29 and significantly increased in 4 of the 156 study catchments. (A large number of catchments showed insignificant decreases, probably due to the large interannual variations of the annual runoff coefficients).

*Despite these issues, I have just three comments on improving the paper:*

*1. The title is misleading. Almost every model will predict discharge changes in response to climate change. The questions is why they do not 'accurately' predict discharge changes? The addition of a qualifier like 'accurately' would be useful.*

> We will revise the title and add 'correctly' as a qualifier. The title then reads 'Why does a conceptual hydrological model fail to correctly predict discharge changes in response to climate change?'.

*2. Changes in anthropogenic influences are largely ignored as the authors claim that the catchments are largely unregulated and existing diversions were introduced before the beginning of the study period. I would question this. While the diversions may be in place before the beginning of the study period, are there operating rules related to this diversions which may vary from year to year, for example allowing larger diversions during periods of low flow (or vice-versa). I ask as we have identified catchments in Australia that not only behaved abnormally (gave lower than predicted yields during the Millennium drought), but that have not returned to 'normal' yields post-drought. One hypothesis for this is that farmers sank groundwater bores to access an alternative water supply during the drought when they were unable to pump from surface water. Any lowering of the groundwater table resulting from this activity would obviously lead to lower than expected yields. Once this 'sunk cost' had been incurred, there would be no benefit to farmers in ceasing the pumping of water from these bores, thus they may still be doing so post-drought. Such anthropogenic influences are of course hard to determine (and even harder to quantify), but the authors would do well to keep them in mind.*

> Changes in private abstractions by households or farmers are indeed difficult to get hold of. In Austria, water abstractions for irrigation are much less important than in Australia. With respect to anthropogenic impacts on water resources, diversions for hydropower generation are much more relevant than abstractions for irrigation. Irrigation in agriculture is most relevant in small areas east, southeast and northwest of Vienna, where estimated irrigation amounts of agricultural areas exceed 10 mm/year (BMLFUW, 2011). In most parts of Austria, estimated irrigation amounts of agricultural areas are less than 1 mm/year. The fraction of arable land in our study catchments is only small (5% on average over the catchments) and the catchments in our study hardly overlap with those areas where agricultural areas receive a large amount of irrigation. At this stage we therefore assume that changes in irrigation amounts are not a major source for the deviations between the simulated and observed discharge changes.

*3. The assessment that problems with the model calibration can be the source of the poor performance during the evaluation period is a good one. In particular, that processes that are relevant in the calibration period are not present (or 'activated' to use the author's terminology) in the calibration period. I am not sure that extending the calibration period from 5 to 25 years will actually evaluate whether this is the case. It may be that these processes will be seen in the 25 year period, but it may not. One thing that could be done is to compare the model that is calibrated on the evaluation period (or perhaps part of it) to the model that is calibrated on the calibration period. If different processes are dominant in the evaluation period, this would be seen in how these models perform on an independent data set.*

In the original version of the model, the model is calibrated in a 5-year period and then evaluated in 6 other 5-year periods. For example, the model calibrated in 1978–1982, is evaluated in 1983–1987, 1988–1992 and so on. If this model performs well in calibration (e.g. in 1978–1982) but performance is worse in evaluation (e.g. 1983-1987), this might be due to a process that was seen in the evaluation but not in the calibration period. If the model is calibrated over a 25 yrs period that includes both 1978-1982 and 1983-1987, the process that was relevant only in 1983–1987 is now included in the calibration period. If there are potentially additional processes that are however not seen in the 25-year calibration period, these processes cannot explain the decrease in model performance when e.g. calibrating in the first 5 yrs of this period and evaluating over the other 20 yrs of this period.

**Replies to Chang Liao**

We would like to thank Chang Liao for his interest in our paper and for uploading his comments.

Below, comments by Chang Liao are in italic font and our replies are in normal font.

*This manuscript tries to untangle one of the most challenging problems in hydrology, and it has implications to more than hydrology models: why even a calibrated hydrology model is not reliable for future simulations? While the authors lay down quite great effort to test and examine some hypothesis, its vision and credibility may be shorten by some major limitations. It is great to see authors went through input driving data (precipitation, temperature, etc.) to all the way up to discharge. The whole analytical process was very convincing. Regardless of the model details, I only have a couple of concerns and comments.*

*First, various spatially distributed hydrology models were used across scales. The authors need to justify why HBV is representative here. There are models considering vegetation dynamics for example.*

We are not claiming that the applied model is representative for all hydrological models and acknowledge that there are models that consider vegetation dynamics. However, conceptual HBV-type models are often used in the context of national scale climate change impact assessments. The fact that in this study, HBV did not result in reliable discharge simulations in a transient climate is thus concerning and very relevant for studies that apply HBV-type models (or similar models that neglect changes in vegetation dynamics).

*Second, as authors pointed out many sources may contribute to model low performance, I suggest there should be at least more evaluations of various hydrological processes. For example, the spatial maps of snow cover, SWE, canopy interception, runoff, snowmelt, soil moisture, etc. A cost function only focus on discharge will likely miss a lot of information. We all know a combination of different parameters can produce the similar results but only one of them is the correct set. The only way to reduce this uncertainty is to examine every single step.*

We agree that including more data on other variables than discharge in the objective function is a good idea. However, for most of the suggested fluxes or state variables there are no observations to compare to (or, available observations are not directly comparable to the modelled variable, as for example for remotely sensed soil moisture). Since many of the study catchments are in a mountainous region, snow data are a relevant data source and we will add a model variant where snow data are included in the objective function (model variant V6, see Tables 2 and 3). The results are described in section 3.3.2.

"Including a snow related criterion into the objective function (model variant V6) improved the model performance with respect to snow without deteriorating the model performance for discharge (Supplementary Table S1). The performance of the model compared to observed snow cover derived from interpolated snow depth was comparable to Parajka et al. (2007), when considering the same set of catchments. Model performance with respect to long-term trends was not improved, with an average gap between simulated and observed discharge trends of $91 \pm 50$ mm $yr^{-1}$ per 35 yrs over 1978–2013 (Table 4)."
* * *
**Replies to comments by Yan Liu et al.**

We would like to thank Yan Liu, Veit Blauhut, Amelie Herzog, Tunde Olarinoye and Ruth Stephan for their interest in our paper and for posting their comments on our manuscript.

Below, their comments are in italic font and our replies are in normal font.

*Comments are from the discussion during a workshop by: Yan Liu, Veit Blauhut, Amelie Herzog, Tunde Olarinoye, Ruth Stephan*

*The study "Why does a conceptual hydrological model fail to predict discharge changes in response to climate change?" by Duethmann et al presents a very interesting topic, which tries to find important factors that influence the prediction capability of conceptual hydrological models, especially under climate change. In this study, the HBV model was used as one representative of conceptual hydrological models. Three aspects regarding precipitation input, model calibration period, and potential evapotransporation (LAI and NDVI were used to consider changes of vegetation dynamics and land cover) were investigated to discuss the causes why HBV model fails to predict discharge under changing climate. This study is in the scope of HESS and well written.*

*After reading and discussing this manuscript during a workshop, we thought that posting our comments might be helpful for improving the manuscript. We have following major and specific points:*

*Major points: 1) Title and abstract are a bit misleading because the results are not generalising for all hydrological models but using HBV as one representative. It would be better to explicitly state that the results are based on HBV model in the abstract. Using subtitle may also help clarifying this issue.*

We will revise the title, please also refer to the comment #1 by David Post. The new title reads 'Why does a conceptual hydrological model fail to correctly predict discharge changes in response to climate change?'. By referring to 'a conceptual hydrological model' and not 'conceptual hydrological models' we intend to indicate that we have tested one and not several or more models. We will make the abstract clearer and mention explicitly that the results are based on a HBV-type model (and catchments in Austria). To avoid abbreviations in the title we did not add 'a HBV-type model' there. Our results are based on a specific model and catchments in Austria, the problems we found may, however, also be relevant for other hydrological models and other regions (also see SC3 by Taehee Hwang).

*2) The prior distribution of model parameters was assumed to be the beta-distribution. In such way, by giving shaping parameters _ and _ for the beta distribution, it seems that the optimal parameter ranges (high probability density part of the beta distribution) are known for the prior. That will affect the model calibration. To justify why using a beta distribution not a uniform distribution for the parameter prior distribution is needed in the method section.*

The a priori distributions for the model parameters were applied to be consistent with the study by Merz2011. It is assumed that we have more information on the likely parameter values than just the parameter range. We checked that including the penalty for deviating from the prior distributions does not have much influence on changes in model performance over time (see Figure 1 below, compared to Figure 3 in the manuscript). When the penalty for deviating from the prior distributions was omitted from the objective function, calibrating the model in subperiod S1 and applying it to 1978–2013 resulted in an average discharge trend of $118 \pm 86$ mm yr$^{-1}$ per 35 yrs and thus virtually no effect compared to the original model.

[Figure]

**Figure 1** (a) Bias and (b) NSE for the different subperiods averaged over all study catchments when omitting the penalty for deviating from the prior distributions. Each line refers to models calibrated in one subperiod, showing bias and NSE during calibration (marked by the filled circle) and during evaluation in the other six subperiods.

*3) Since the results were analyzed for the averages over 156 catchments, it would be better to see the probability density distribution of the bias (Qobs-Qsim) of all catchments for the prediction periods to support that the low predictability of the averages of all catchments is not due to several catchments that bring very big bias. Providing this information in the supplement will strongly support the results.*

This is a good idea and we will add violin plots showing the distribution of the bias and NSE as Supplementary Figures S4 and S5. Figure S4 shows that the changes in the average bias were not caused by few catchments with very large changes.

*Specific points:*

*1) A northern arrow is missing in Fig.1, the elevation legend is normally vertical. Fig. 2 is not very informative, maybe merge it with Fig. 1.*

A northern arrow will be added. The elevation legend was set horizontal to better use the space. Figure 2 will be moved to the supplement.

*2) In Fig. 4, how was the bias calculated.*

This will be added. The bias was calculated as $bias = \left(\sum_{t=1}^{n} Q_{sim,t} - \sum_{t=1}^{n} Q_{obs,t}\right) / \sum_{t=1}^{n} Q_{obs,t}$

*3) What does the unit "mm yr-1 per 35 yrs" mean? Is that the mean discharge (mm yr-1) over the 35 years?*

The unit "mm yr-1 per 35 yrs" refers to trends, such as the trend in mean annual discharge over a period of 35 yrs.

*4) In equation 8, definition of fbeta is missing. fp was not used.*

Thanks, this will be corrected.

*5) In Sect. 2.3.1, many model parameters were introduced, such as CR and Bmax, but these two parameters are not provided in Table 1.*

Table 1 lists parameter ranges of the a priori distribution for the model parameters that were included in the model calibration. The parameters $T_R$, $T_S$, $C_r$ and $B_{max}$ were not included in the model calibration and set to constant values (see section 2.3.3). They are therefore not listed in Table 1 and we will add a not to the table header to clarify this.

*6) Table 2 contains almost all the details of hypotheses. But there is also quite long text in Sect. 2.4.2 that repeats the table. Table 2 is clear, try to reduce the duplicate text in Sect. 2.4.2.*

We include small changes in 2.4.2. However, since the text contains further explanations of the hypotheses that are summarized in Table 2, large parts were left as in the original manuscript.

*7) Hypothesis should be a result out of the introduction and be mentioned at the last paragraph in the introduction.*

That is an alternative we have also thought about. However, the reason why we decided to introduce the detailed hypotheses after the model description and not in the introduction was

that we assume it to be easier for a reader to follow the hypotheses and the corresponding modifications to the model after the model (including the input data) has been introduced.

*8) In the discussion, very good literature review was done. But it should more highlight the findings of this study and relate and compare to literatures.*

We modified the discussion, also in response to a comment by Referee #2. We discuss the results of this study in the context of existing studies on this topic.

*9) It is not clear that how the trend was calculated when using 25 years as the calibration period. Please clarify that.*

As in the cases where the model is calibrated for 5-year periods, the parameters are applied to the entire study period and the trend over the entire study period is calculated.
* * *
**Reply to Taehee Hwang**

We thank Taehee Hwang for his entirely positive comment on our paper. Taehee Hwang replied to a comment by Dr. Liu. He underlined the relevance of our study by emphasizing the problem that many hydrologic models neglect changes in vegetation dynamics even though vegetation responses to climate change may have important influences on hydrologic systems.

[revised manuscript text omitted]

Supplement

**Why does a conceptual hydrological model fail to correctly predict discharge changes in response to climate change?**

Doris Duethmann[1, 2], Günter Blöschl[1], Juraj Parajka[1]

[1]Institute for Hydraulic and Water Resources Engineering, Vienna University of Technology, Karlsplatz 13/223, 1040 Vienna, Austria.
[2]IGB Leibniz Institute of Freshwater Ecology and Inland Fisheries, Müggelseedamm 310, 12587 Berlin, Germany.

*Correspondence to*: Doris Duethmann (duethmann@igb-berlin.de)

**Supplement S1 Influence of including a penalty for model parameters that deviate from an a priori distribution into the objective function**

The objective function applied for model calibration contains a penalty for model parameters that deviate from an a priori distribution, consistent with the study by Merz2011. In order to test the possible influence of this criterion on the difference between simulated and observed discharge trends, we also performed simulations where the model was calibrated without this criterion, i.e. $w_3$ in Eq. 1 was set to 0. This had only small effects on changes in model performance over time (Fig. S1). When the penalty for deviating from the prior distributions was omitted from the objective function, calibrating the model in S1 and applying it to 1978–2013 resulted in a gap between simulated and observed discharge trends of $88 \pm 48$ mm yr$^{-1}$ per 35 yrs and thus negligible differences to the original model with a gap of $92 \pm 50$ mm yr$^{-1}$ per 35 yrs.

[Figure]

**Figure S1** (a) Bias and (b) NSE for the different subperiods averaged over all study catchments when omitting the penalty for deviating from the prior distributions. Each line refers to models calibrated in one subperiod, showing bias and NSE during calibration (marked by the filled circle) and during evaluation in the other six subperiods.

**Further supplementary tables and figures**

**Table S1** Model performance with respect to discharge and snow cover for the baseline model V0 and model variant V6 (where a criterion on snow cover was included in the objective function), as averages over all catchments. Values indicate the range over 7 calibration periods and 42 evaluation periods (using the other 6 subperiods for each calibration period). NSE: Nash-Sutcliffe efficiency for discharge, bias: volume bias for discharge, $Z_s$: ratio of days with poor snow cover performance (see main text) to the total number of days in the simulation period.

|  | NSE | | Bias | | $Z_s$ | |
|---|---|---|---|---|---|---|
|  | calibration | evaluation | calibration | evaluation | calibration | evaluation |
| V0 baseline model | 0.70–0.75 | 0.56–0.71 | 0.005–0.03 | -0.13–0.18 | 0.08–0.12 | 0.07–0.13 |
| V6 include snow data in calibration | 0.70–0.75 | 0.58–0.71 | 0.004–0.04 | -0.11–0.18 | 0.05–0.08 | 0.05–0.09 |

[Figure]

**Figure S2** Number of precipitation and air temperature stations included for the interpolation of precipitation and air temperature in the data sets P0 and T0.

[Figure]

**Figure S3** Temporal variations in simulated and observed water balance derived evaporation plus storage changes (calculated as precipitation minus simulated respectively observed discharge), as averages over all study catchments. The thick lines show subperiod annual means, the thin lines annual sums, and the dashed lines linear trends.

[Figure]

**Figure S4** Violin plots showing the distribution of the bias across the 156 study catchments of the models calibrated in subperiod S1 (a), S2 (b), S3 (c), S4 (d), S5 (e), S6 (f), and S7 (g), evaluated for subperiod S1–S7 (*x*-axis in each subplot). Grey crosses represent the mean and standard deviation. The means are the same as shown in Fig. 4 (a).

[Figure]

**Figure S5** Violin plots showing the distribution of NSE across the 156 study catchments of the models calibrated in subperiod S1 (a), S2 (b), S3 (c), S4 (d), S5 (e), S6 (f), and S7 (g), evaluated for subperiod S1–S7 (*x*-axis in each subplot). Grey crosses represent the mean and standard deviation. The means are the same as shown in Fig. 4 (b).

[Figure]

**Figure S6** Spatial patterns of trends of the differences between simulated discharge for selected model variants and **(a–c)** observed discharge, or **(d–f)** simulated discharge of the baseline model V0. **(a,d)** refer to model variant V2, **(b,e)** to V7, and **(c,f)** to V8. Filled circles indicate significant trends at $p{\leq}0.05$.

---

## Author Response (AR2)

Dear Editor,

Thank you for carefully reading. All suggested corrections were included as detailed below.

During final checks to the paper and applied code, we found an error that affects the calculation of the trend slopes. In the previous version, an arithmetic mean was used for the calculation of Sen's slope where it should be the median (which had not been changed back after a test). This has been corrected, and we have updated the respective numbers in the text, figures, and tables. The resulting changes in the trend estimates were generally small. Furthermore, figures S4 and S5 in the Supplement were updated using a different option for smoothing the histogram, which better reflects the very narrow distributions of bias during the calibration period.

Kind regards,

Doris Duethmann and co-authors

**Comments by the Editor**

*- p. 7, l. 21, the reference Tucker et al. (2005) is missing in the list of the references, please, add it*

> Thanks, done.

*- p. 8, l. 10, could you add a comment why have you chosen the multiplication factor of 1.2 for E2. Is this simply the value of the annual average ratio of E2 to E1 averaged over all catchments, maybe you could add an explanation that this value is not arbitrary (based on) but exactly this ratio*

> Ok. The sentence was corrected as follows "In order to avoid water balance problems in the hydrological model, E2 was multiplied with the annual average ratio of E2 to E0 averaged over all catchments with a value of 1.2."

*- p. 20, l. 12 to l. 16, please check the availability of data and change the last access date to the current one for three web pages*

> Done.

*- p. 21, l. 31, the year should be the last information, please, type as ", 012006, 2010."*

> Done.

*- it seems that you do not refer to Figure 1 in the main text, please, add*

> Thanks, this was added.

[revised manuscript text omitted]